# Using Seismic Attributes in seismotectonic research: an application to the Norcia's Mw=6.5 earthquake (30th October 2016) in Central Italy.

Maurizio Ercoli[1;4], Emanuele Forte[2], Massimiliano Porreca[1;4], Ramon Carbonell[3], Cristina Pauselli[1;4], Giorgio Minelli[1;4], Massimiliano R. Barchi[1;4].

[1] Dip. di Fisica e Geologia – Università degli Studi di Perugia (Perugia, Italy).
[2] Dept. of Mathematics and Geosciences, University of Trieste (Trieste, Italy).
[3] Dept. Structure & Dynamics of the Earth, CSIC-Inst. Earth Sciences Jaume Almera (Barcelona, Spain).
[4] Member of Interuniversity Center for Research on 3D-Seismotectonics (Centro InterRUniversitario per l'Analisi SismoTettonica tridimensionale con applicazioni territoriali – CRUST).

*Correspondence to*: Maurizio Ercoli (maurizio.ercoli@unipg.it; maurizio.ercoli@gmail.com)

**Abstract.** In seismotectonic studies, seismic reflection data are a powerful tool to unravel the complex deep architecture of active faults. Such tectonic structures are usually mapped at the surface through traditional geological surveying, whilst seismic reflection data may help to trace their continuation from the near-surface down to hypocentral depths. On seismic reflection data, seismic attributes are commonly used by oil and gas industry to aid exploration. In this study, we propose to use seismic attributes to seismotectonic research for the first time. The study area is a geologically complex region of Central Italy, struck during the 2016-2017 by a long-lasting seismic sequence, including a Mw 6.5 main-shock. Three vintage seismic reflection profiles are currently the only ones available at the regional scale across the epicentral zone. These represent a singular opportunity to attempt a seismic attribute analysis, by running attributes like the "Energy" and the "Pseudo Relief". Our results are critical, as they provide information on the relatively deep structural setting, mapping a prominent, high amplitude regional reflector interpreted as the top of basement, which is an important rheological boundary. Complex patterns of high-angle discontinuities crossing the reflectors have also been identified by seismic attributes. These steeply dipping fabrics are interpreted as the expression of fault zones, belonging to the active normal fault systems responsible for the seismicity of the region. Such peculiar seismic signatures of faulting are consistent with the principal geological and tectonic structures exposed at surface. In addition, we also provide convincing evidence of an important primary tectonic structure currently debated in the literature (the Norcia antithetic fault) as well as several buried secondary fault splays. This work demonstrates that seismic attribute analysis, even if used on low-quality vintage 2D data, may contribute to improve the subsurface geological interpretation in areas characterized by limited and/or low-quality subsurface data but with potentially high seismic hazard.

# 1 Introduction

Studying the connections between the earthquakes and the faults to which they are associated is a primary goal of seismotectonics (Allen et al., 1965; Schwartz and Coppersmith, 1984). Within this field, it is generally complex filling the gap between the exposed geology (including the active "geological faults") and the seismological data (e.g. focal mechanisms, earthquake locations, etc...), indicators of the geometry and kinematics of the seismic source at hypocentral depth ("seismological faults", sensu Barchi & Mirabella, 2008). The recovery of information on the seismogenic structures at depth is difficult, primarily due to the lack of high-resolution geophysical data and/or well stratigraphy. The lack of these data results in relatively high degrees of uncertainty, and drives to contrasting geological models and interpretations. Different geophysical methods (e.g. active/passive Seismic, Gravimetry, Magnetics, Electric and Electromagnetic such as Magnetotellurics and Ground Penetrating Radar) may contribute to define the stratigraphy and structural setting of the upper crust at different scales. The data provided by seismic reflection technique are poorly affected by well-known inversion problems typical of the potential methods (Snieder & Trampert, 1999), and are largely the most powerful tool able to produce high-resolution images of the subsurface. This type of data, if calibrated with deep well's stratigraphy provides very strong constraints to the definition of subsurface geological architecture. These profiles are useful to unveil the deep geometry of active faults mapped in the field, and extend them down to hypocentral depths. Unfortunately, ex-novo acquisition (possibly 3D) of onshore deep seismic reflection data for research purposes, is hampered by high costs, environmental problems and complex logistics (e.g. prohibition of dynamite or vibroseis trucks in Natural Parks or urban areas). Significant exceptions are research projects for deep crustal investigations like BIRPS (Brewer et al., 1983), CoCORP (Cook et al., 1979), ECORS (Roure et al., 1989) and CROP (Barchi et al., 1998; Finetti et al., 2001), IBERSEIS (Simancas et al., 2003), ALCUDIA (Ehsan et al., 2014 and 2015) among others. In seismically active regions, old profiles (legacy data) acquired by the industry have been successfully used, to connect the active faults mapped at the surface with the earthquakes seismogenic sources depicted by seismological records (Boncio et al., 2000; Bonini et al., 2014; Carvalho et al., 2008; Beidinger et al., 2011; Maesano et al., 2015; Porreca et al., 2018). Legacy seismic lines have in fact some advantages: 1) they are already available from the oil companies, national archives 2) they represent a nice source of information in places where new data is difficult to acquire; 3) they can be used to build up and refine geological models. Moreover, such data are often the only one available. Therefore, this legacy data is very valuable and it's worth to use them to constrain the subsurface geological setting and, to provide new data on active tectonic structures (see DISS database, Basili et al., 2008). Vintage profiles can therefore significantly contribute to seismo-tectonic researches, even if characterized by intrinsic limitations: i) their location, orientation and acquisition parameters were not specifically designed for this aim; ii) they were collected using relatively old seismic technologies and acquisition/processing strategies. Consequently, these produced data with relatively low signal/noise ratio (S/N) and low resolution, especially when compared to modern standards (Manning et al., 2019). In order to improve the image quality and increase the accuracy of the interpretation, two main strategies,

ordinarily used by the O&G industry, can be applied to legacy data: 1) reprocessing from raw data using modern processing
strategies and newly designed algorithms and software; 2) use post-stack analysis techniques such as seismic attributes.
An attribute analysis is, perhaps, one of the easiest, cheapest and fastest strategies to qualitatively emphasize the geophysical
features and data properties of reflection seismic data sets, producing benefits particularly in complex geological areas. A
seismic attribute is a quantity derived from seismic data (pre-stack and/or post-stack) commonly used to extract additional
information that may be unclear in conventionally processed seismic lines. Examples of applications on dense 3D seismic
volumes produced impressive results, including: the identification of ancient river channels; or, sets of faults at variable
scales (Chopra & Marfurt, 2005; Chopra & Marfurt, 2007; Chopra & Marfurt, 2008; Marfurt et al., 2011; Hale, 2013;
Barnes, 2016, Iacopini et al., 2016; Marfurt, 2018; Wrona et al., 2018; Di & AlRegib, 2019; Naeini & Prindle, 2019).
There are several advantages in using 3D seismic data instead of 2D. Advantages of 3D and pitfalls of 2D are extensively
discussed in Torvela et al. (2013) and Hutchinson (2016). 2D seismic data is more limited and 2D post-stack seismic
attribute analysis may not provide the same quality of information as when using 3D (Marfurt & Alves, 2015; Ha et al.,
2019). However, the main point is that in the past, it was common to sample study areas inland by 2D grids of seismic
profiles, being the full 3D seismic surveys rare.  Hence, it is relevant to extract as much information as possible from 2D
data.
In this work, the selected study area is located between the southeastern part of the Umbria-Marche Apennines and the Laga
Domain, in the outer Northern Apennines (central Italy) (e.g. Barchi et al., 2001). This area presents ideal characteristics to
test the application of seismic attributes as a new approach in seismotectonics. In the past, several seismic profiles were
acquired in this region for hydrocarbon exploration, and were later used to constrain subsurface geological structures (Bally
et al., 1986; Barchi et al, 1991; Barchi et al., 1998; Ciaccio et al., 2005; Pauselli et al., 2006; Mirabella et al. 2008; Barchi et
al., 2009; Bigi et al., 2011). After the 2016-2017 seismic sequence, Porreca et al. (2018) provided an updated regional
geological model based on the interpretation of vintage seismic lines. However remarkable differences in the seismic data
quality across the region, prevented a straightforward seismic interpretation. Therefore, the present work exploits the use of
seismic attributes on three low-quality seismic profiles located close to the Mw 6.5 main-shock of the 2016-2017 seismic
sequence. The main goal is to squeeze additional information from the 2D data obtaining as many constraints as possible on
the geological structures responsible for the seismicity in the area, by defining:
- geological/structural setting at depth (e.g. depth of the basement and its involvement)
- trace of potentially seismogenic faults (connection between the active faults mapped at the surface and earthquake's foci).
Any improvements achievable on the data quality and visualization, for example an increase of the resolution and/or an
enhancement of the lateral extent or limits of the seismic reflectors, would represent a valuable contribution considering the
limited amount of data available in this area. We think that this innovative approach to seismotectonic research can be
extended to other on-shore seismically active areas in the world, especially if covered only by sparse vintage low-quality
seismic surveys. In such cases, we think the seismotectonic research may benefit of the potential and improvements
generated by the seismic attributes.

## 2 Geological framework and seismotectonics of the study area

The study area is located in the southeastern part of the Northern Apennines fold and thrust belt. The area includes the Umbria-Marche Domain and the Laga Domain, which are separated by an important regional tectonic structure, known as the M. Sibillini thrust (MSt) (Fig. 1). The Umbria-Marche domain involves the rocks of the sedimentary cover, represented by three main units (top to bottom), characterized by different interval velocities (Bally et al., 1986; Barchi et al., 1998; Porreca et al., 2018):

1) on top, the Laga sequence (Late Messinian – Lower Pliocene, up to 3000 m thick, average seismic velocity; $V_{av}$ = 4000 m/s). It consists of siliciclastic turbidites made by alternating layers of sandstones, marls and evaporites, deposited in marine depositional environment (Milli et al., 2007; Bigi et al., 2011); it outcrops in the eastern sector of the study area (i.e. Laga Domain);

2) the carbonate formations (Jurassic-Oligocene, about 2000 m thick, $V_{av}$ = 5800 m/s), formed by pelagic limestones (Mirabella et al., 2008) with subordinated marly levels overlying an early Jurassic carbonate platform (Calcare Massiccio Fm.). It outcrops mainly in the Umbria-Marche Domain;

3) at the bottom the Late Triassic evaporites (1500–2500 m thick, $V_{av}$= 6400 m/s). They consist in alternated layers of anhydrites and dolomites (Anidriti di Burano Fm. and and Raethavicula Contorta beds; Martinis & Pieri, 1964), never outcropping and intercepted only by deep wells (Porreca et al., 2018 and references therein).

For further details on the stratigraphic characteristics of the area, the reader can refer to the works by Centamore et al. (1992) and Pierantoni et al. (2013).

These units rest on a basement with variable lithology (Permian-Late Triassic, $V_{av}$ = 5100 m/s) that never crops out in the study area (Vai, 2001). It has only been intercepted by deep wells (Bally et al., 1986; Minelli & Menichetti, 1990; Anelli et al., 1994; Patacca & Scandone, 2001).

This sedimentary sequence is involved in the Late Miocene fold and thrust belt including a set of N-S trending anticlines, formed at the hangingwall of the W-dipping arc shaped major thrusts. The most important compressional structure is the M. Sibillini thrust (MSt, Koopman, 1983; Lavecchia, 1985), where the Umbria-Marche Domain is overthrusted on the Laga Domain.

This is a geologically complex region, where in the past the analysis of 2D seismic profiles have produced contrasting interpretations of the upper crust structural setting, i.e. thin- vs. thick-skinned tectonics, fault reactivation/inversion and basement depth (Bally et al., 1986; Barchi, 1991; Barchi et al., 2001; Bigi et al., 2011; Calamita et al., 2012). A review of the geological history of this area has recently been provided by Porreca et al. (2018). These authors propose a tectonic style characterized by coexistence of thick- and thin-skinned tectonics with multiple detachments localized at different structural levels.

These compressional structures have been later disrupted by the extensional faults since the Late Pliocene (Fig.1) (Blumetti et al., 1993; Boncio et al., 1998; Brozzetti & Lavecchia, 1994; Calamita & Pizzi, 1994; Pierantoni et al., 2013).

The Late Pliocene-Quaternary extensional tectonic phase, characterized by NNW-SSE striking normal faults, is consistent with the present-day active strain field as deduced by geodetic data (e.g. Anderlini et al., 2016). The latter faults have high dip angles (50-70°) and can be synthetic or antithetic normal structures (WSW or ENE dipping, respectively). These faults were also responsible of the tectono-sedimentary evolution of intra-mountain continental basins (Calamita et al., 1994; Cavinato and De Celles, 1999). The most evident Quaternary basins of this part of the Apennines are the Castelluccio di Norcia and Norcia basins (Fig.1), located at 1270 and 700 m a.s.l., here named CNb and Nb respectively. A phase of lacustrine and fluvial sedimentation infilled both basins with hundred meters of deposits, characterized by fine clayey to coarse grained material (Blumetti et al., 1993; Coltorti and Farabollini, 1995).

The area is affected by frequent moderate magnitude earthquakes (5 < Mw < 7) and has a high seismogenic potential revealed by both historical and instrumental data (e.g. Barchi et al., 2000; Boncio and Lavecchia, 2000; Basili et al., 2008; Rovida et al., 2016; DISS Working Group, 2018). The major seismogenic structures recognized in the area are the Norcia fault (Nf) and the M. Vettore fault (Vf). The Norcia fault (Nf, Fig.1) is associated to several historical events (Galli et al., 2015; Pauselli et al., 2010; Rovida et al., 2016), probably including the 1979 earthquake (Nottoria-Preci fault, Deschamps et al., 1984; Brozzetti & Lavecchia, 1994; Rovida et al., 2016) and, the largest event in 1703 (Me = 6.8, Rovida et al., 2016). The Vettore fault (Vf) is part of the easternmost alignment whose historical and pre-historical activity was recognized by paleoseismological and shallow geophysical surveys (Galadini & Galli, 2003; Galli et al., 2008; Ercoli et al., 2013; Ercoli et al., 2014; Galadini et al., 2018; Galli et al., 2018; Cinti et al, 2019; Galli et al., 2019). This system was reactivated during the 2016-2017 sequence, characterized by multi-fault ruptures occurred within few months (nine M>5 earthquakes at hypocentral depth < 12 km between August 2016 – January 2017) having characteristics comparable to previous seismic sequences in Central Italy (e.g. L'Aquila 2009 and Colfiorito 1997-1998, Valoroso et al., 2013 and Chiaraluce et al., 2005). The strongest mainshock of (Mw 6.5) occurred on 30th October 2016 (Chiaraluce et al., 2017; Chiarabba et al., 2018; Gruppo di Lavoro Sequenza Centro Italia, 2019; Improta et al., 2019; ISIDe working group, 2019), generating up to 2 m (vertical offset) co-seismic ruptures (Civico et al., 2018; Gori et al., 2018; Villani et al., 2018a; Brozzetti et al., 2019), mainly localized along the Mt. Vettore fault (blue thin lines in Fig. 1).

Despite of the large amount of surface data collected (Livio et al., 2016; Pucci at al., 2017; Wilkinson et al., 2017; De Guidi et al., 2017; Brozzetti et al., 2019), the deep extension of the Norcia and Castelluccio antithetic and synthetic faults (particularly Nf and Vf), and the overall complex structure of the area are still debated (Lavecchia et al., 2016; Porreca et al., 2018; Bonini et al., 2019, Cheloni et al., 2018, Improta et al. 2019, Di Giulio et al., 2020) and remains an open question.

**3 Data**

We have performed seismic attributes analysis on three W-E trending 2D seismic reflection data crossing the epicentral area between the Umbria and Marche regions (Central Italy, Fig.1). These seismic profiles are part of a much larger, unpublished dataset including 97 seismic profiles and, a few boreholes, drilled for hydrocarbon exploration by ENI in the period 1970-1998. The data quality is extremely variable (medium/poor) with limited fold (generally < 60 traces / Common Mid-Point),

mainly due to environmental and logistical factors. Among the latter, we can list: the different acquisition technologies; a limited site access; the complex tectonic setting and, especially, the different (and contrasting) outcropping lithologies (e.g. Mazzotti et al., 2000, Mirabella et al., 2008). The eastern area, showing higher data quality, consists of siliciclastic units of the Laga foredeep sequence, located at the footwall of the MSt. On the other hand, the lowest S/N recordings coincide with outcropping carbonates formations and Quaternary deposits.

The analysed lines include seismic reflection profiles NOR01 (stack, 14 km long) and NOR02 (time-migrated, 20 km long, partially parallel to NOR01 on the western sector) located west and east to the Nb, respectively; CAS01 (stack, 16 km long), located further to the south crossing the Cascia village (Fig. 1).

NOR01 and CAS01 were acquired using a Vibroseis source, while explosives were used for NOR02; all the lines are displayed in Two-Way-Travel-Time (TWT) limited to 4.5 s. The amplitude/frequency spectra (computed on the entire time window) of the processed lines show a bandwidth in a range of 10-50 Hz, with the NOR02 spectrum displaying a slightly higher frequency content (Tab.1). Assuming an average peak frequency of 20 Hz, a vertical resolution of ca. 80 m can be estimated (using an average carbonate velocity = 6 km/s; parameters in Table 1). Some processing artefacts are visible in NOR01 as a horizontal signal at ca. 1 s (yellow dashed line and label A in Fig. 2a), and another in CAS01 (Fig. 3a). As suggested in the introduction, we considered that the interpretation could benefit from the application of seismic attributes to the seismic images. However, different sets of parameters need to be tested to achieve relevant improvements. Therefore, we loaded the profiles into the software OpendTect (OdT, https://www.dgbes.com/index.php/software#free). A common seismic datum of 500 m was considered for the transect. Unfortunately, deep borehole stratigraphy is not available for the study area (all details about surrounding deep wells have been already summarized in Porreca et al., 2018). The OdT seismic project was enriched also by ancillary data, extracted by a complementary GIS project (QGis, https://www.qgis.org/it/site/). As visible in Fig. 1, we have included a detailed summary of the main normal faults and surface ruptures of the area (Civico et al., 2018; Villani et al., 2018a; Brozzetti et al., 2019), obtained after carefully checking the most important regional geological maps and fault patterns (Koopman, 1983; Centamore et al., 1993; Pierantoni et al., 2013; Carta Geologica Regionale 1:10'000 – Regione Marche, 2014; Carta Geologica Regionale 1:10'000 – Regione Umbria, 2016; Ithaca database, http://www.isprambiente.gov.it/it/progetti/suolo-e-territorio-1/ithaca-catalogo-delle-faglie-capaci;), as well as the most recent works published in literature (e.g. Brozzetti et al., 2019; Porreca et al., 2020). The topography was also included using a regional 10 meters resolution DTM data base (Tarquini et al., 2007; Tarquini et al., 2012). The other important external data-set consists of seismological data, i.e. inferred location and approximated fault geometry as suggested by the focal mechanisms of the mainshocks and, by the distribution of the aftershocks (Iside database, http://iside.rm.ingv.it/iside/ and Chiaraluce et al., 2017). The integration of such information in a pseudo-3D environment offered us a multidisciplinary platform to clearly display the seismic lines and establish links between surface data the interpreted deep geologic structures located at hypocentral depths.

## 4 Methods

The seismic reflection data interpretation is generally accomplished by correlating specific signal characteristics (seismic signature), with the different geological domains identified within the study area. A standard seismic interpretation is affected by a certain degree of uncertainty/subjectivity (particularly in case of poor data quality), because is generally based on a qualitative analysis of amplitude, geometry and lateral continuity of the reflections. Over the last years, the introduction of seismic attributes and related automated/semi-automated procedures has had an important role in reducing the subjectivity of seismic interpretations and achieve quantitative results. A seismic attribute is a descriptive and quantifiable parameter that can be calculated on a single trace, on multiple traces, or 3D volumes and can be displayed at the same scale as the original data. Seismic data can be, therefore, considered a composition of constituent attributes (Barnes, 1999, Taner et al., 1979, Forte et al., 2012). Their benefits have been first appreciated in 2D/3D seismic reflection data (Barnes 1996; Taner et al., 1979; Barnes, 1999; Chen and Sidney, 1997; Taner, 2001; Chopra and Marfurt, 2007; Chopra and Marfurt, 2008; Iacopini and Butler, 2011; Iacopini et al., 2012; McArdle et al., 2014; Botter et al., 2014; Hale, 2013 for a review; Marfurt and Alves, 2015; Forte et al., 2016) and, more recently, also in other subsurface imaging techniques like Ground Penetrating Radar (e.g. McClymont et al., 2008; Forte et al., 2012; Ercoli et al., 2015, Lima et al., 2018). In this work, we have tested several post-stack attributes on three 2D vintage seismic lines (original seismic data in the supplementary material in Fig.1s). We started our analysis by using first the well-known and widely used attributes like the instantaneous amplitude, phase, frequency, and their combinations. We also used composite multi-attribute displays (i.e. simultaneous overlay and display of different attributes e.g. primarily phase, frequency, envelope; Chopra and Marfurt, 2005; Chopra and Marfurt, 2011). Later on, we have also tested other attributes like coherency and similarity, which are generally more efficient on 3D volumes. These did not result in positive outcomes, due to the limited vertical and lateral resolution of our legacy data.  Among the tested attributes, we selected three ones that resulted in the best images (provided in Figs. s2, s3 and s4 of the supplementary material, without any line drawing or labels), aiding the detection of peculiar seismic signatures related to the regional seismogenic layers and fault zones. The attributes, computed using OdT software, are:

**"Energy" (EN):** one of the RMS amplitude-based attributes, it is defined as the ratio between the squared sum of the samples amplitude values in a specified time-gate and the number of samples in the gate (Taner, 1979, Gersztenkorn & Marfurt, 1999, Chopra & Marfurt, 2005, Chopra & Marfurt, 2007, for a review of formulas see Appendix A in Forte et al., 2012). The Energy measures the reflectivity in a specified time-gate, so the higher the Energy, the higher is the reflection amplitude. In comparison to the original seismic amplitude, it is independent of the polarity of the seismic data being always positive, and in turn preventing the zero-crossing problems of the seismic amplitude (Forte et al., 2012, Ercoli et al., 2015, Lima et al., 2018, Zhao et al., 2018). This attribute is useful to emphasize the most reflective zones (e.g. characterization of acoustic properties of rocks). It may also enhance sharp lateral variations in seismic reflectors, highlighting discontinuities like fractures and faults. In this

work, we decided to use a 20 ms time window (i.e. close to the average wavelet length), obtaining considerable
improvements in the visualization of higher acoustic impedance contrasts.
**"Energy gradient" (EG)**: it is the first derivative of the energy with respect to time (or depth). The algorithm
calculates the derivative in moving windows and returns the variation of the calculated energy as a function of time
or depth (Chopra & Marfurt, 2007; Forte et al., 2012). It is a simple and robust attribute, also useful for a detailed
semi-automatic mapping of horizons with a relative low level of subjectivity. The attribute acts as an edge detection
tool. It is effective in the mapping of the reflection patterns as well as the continuity of both steep discontinuities
like faults and fractures, and channels, particularly in slices of 3D data (Chopra & Marfurt, 2007). In this work, we
have selected for a time window of 20 ms. We have obtained considerable improvements in the visualization not
only of the strong acoustic impedance reflectors, but particularly in the faults signature imaged in the shallowest
part of the seismic sections.
**Pseudo-relief (PR):** it is obtained in two steps: the energy attribute is first computed in a short time window, then
followed by the Hilbert transform (phase rotation of -90 degrees). The Pseudo-relief is considered very useful in 2D
seismic interpretation to generate "outcrop-like" images. It allows an easier detection of both faults and horizons
(Bulhões, 1999; Barnes et al., 2011; Vernengo et al. 2017, Lima et al., 2018). In this work, considerable
improvements have been obtained by computing the Pseudo-relief using a window length of 20 ms. In comparison
to the standard amplitude image, it highlights the reflection patterns and thus the continuity/discontinuity of
reflectors, enhancing steep discontinuities and fault zones.

## 5 Results

The Figs. 2, 3 and 4 show the comparison between the original seismic lines in amplitude and, the images obtained after the
attribute analysis, revealing significant improvements in the visualization and interpretability of the geophysical features. In
the profiles NOR01, CAS01 and NOR02 we focus our analysis on three types of geophysical features highlighted by the
attributes: sub-horizontal deep reflectors, low-angle and high-angle discontinuities. The main faults known at the surface
(Fig.1) have been also plotted on top of each seismic line.
In the original seismic line NOR01 (Fig. 2a), the overall low S/N ratio hampers the detection of clear and continuous
reflectors. At ca. 1 s, a horizontal processing artefact is visible (label A, yellow dots), possibly related to a windowed filter.
The most prominent sub-horizontal reflections (labelled H) are located in the central portion between 2-3 s (TWT) (strong
reflectors in the black box i). Shallower and less continuous reflectors are also visible in the eastern side of the profile,
beneath the Nb (black box ii). The EN attribute (Fig. 2b) enhances the reflectivity contrast, better focusing the high-
amplitude, gently W-dipping reflector H (blue arrows) and also outlining its lateral extension. In this image most of the
reflected energy is concentrated on its top, at ca. 2.5 s: it is readily apparent that H separates two seismic facies, with higher
(top) and lower (bottom) amplitude response, respectively. The EG and PR attributes of NOR01 (Figs. 2c, 2d) better display
the geometry of horizon H, characterized by a continuous, ca. 8 km long, package of reflectors (ca. 200 ms thick) having

common characteristics in terms of reflection strength and period. In the eastern part of the profile, below the Nb, the EG and PR attributes also enhance two major opposite-dipping high-angle geophysical features (red arrows in fig. 2c and 2d), crossing and disrupting the shallower reflectors. The W-dipping lineament propagates down to ca. 2.5 s, intercepting the eastern termination of the reflector H. The two discontinuities define a relatively transparent, shallow seismic facies, corresponding to the area where the Nb outcrops. In the same sector, the reflectors are pervasively disrupted by many other, minor discontinuities.

The original seismic reflection line CAS01 (Fig. 3a) displays a generalized high-frequency noise content. As in NOR01, a shallow processing artefact (A, yellow dots) is visible and possibly related to the application of a windowed filter. Fragmented packages of high-amplitude reflectors (H) are visible at the same time interval observed in NOR01 (ca. 2.5 s), in both the western (black box i, in Fig. 3a) and, more discontinuous, in eastern part of the line (black box ii, in Fig. 3a). The EN attribute (Fig. 3b) emphasizes the presence of the H reflector, better focusing its reflectivity (blue arrows). Both the EG and PR attributes (Figs. 3c and 3d) further help to delineate the reflector H. The steeper discontinuities have been analysed mainly in the western part of the profile, closer to the 2016-2017 seismically active area. A major high-angle, east-dipping discontinuity has been traced at about 13 km (alignment of red arrows in Fig. 3c and 3d).

The original seismic line NOR02 (Fig. 4a), displays geophysical features similar to the ones detected in NOR01 and CAS01. This seismic profile shows a generalized poor/limited lateral continuity of the reflectors, with the exception of the eastern side. In this sector, a set of west-dipping coherent reflections can be recognized: the higher S/N ratio correlates with the outcropping turbidites of the Laga sequence, which are known to favour the seismic energy penetration and reflection, in comparison to carbonates (e.g. Bally et al., 1986; Barchi et al., 1998). The prominent reflection H, gently east-dipping and relatively continuous for more than 8 km (black box in Fig 4a), is located in the centre of the line, at greater depth (3.2–3.5 s TWT), respect to the previously described NOR01 and CAS01 profiles. As in the previous cases, the EN attribute (Fig. 4b) effectively focuses the horizons reflectivity, emphasising the high amplitude of the reflector H (blue arrows). The EG and PR attributes (Figs. 4c and 4d) improve the overall visualization of the reflection patterns, aiding the detection of the low-angle and high-angle discontinuities. A major westward low-angle discontinuity T (green dots in Figs.4c and 4d) crosses the entire profile, descending from ca. 2 s (East) to ca. 4 s (West), where it intersects the reflector H. Several high-angle discontinuities have been traced along the section, marked by the alignments of red arrows in Figs. 4c and 4d. The most relevant alignments have been recognised beneath the two major Quaternary basins (i.e. Nb and CNb) crossed by the profile: in both cases, major W-dipping alignments can be traced from the near surface, where they correspond to the eastern border of the above mentioned basins, down to a depth of ca. 4 s TWT. Other discontinuities, W and E dipping, have been traced in the hanging-wall of these two major alignments. In the seismic line sector bounded by these features, many secondary (minor) discontinuities pervasively cross-cut the set of reflectors, producing a densely fragmented pattern. Unfortunately, the limited resolution and data quality in the deeper part of the section hampers a univocal interpretation of the cross-cutting relationships between the low-angle discontinuity T and the W-dipping high-angle discontinuity: two alternative interpretations are possible, that will be discussed in detail in the next paragraph 6.

The global improvement in the dataset interpretability can be better appreciated in a 3D visualization of the seismic
attributes, also using multi-attribute displays (Fig. 5). Such images reveal the deep geometry of the main reflectors and the
location of the geophysical discontinuities, later interpreted in the light of known and debated tectonic structures on the study
area. In Fig. 5a we report a 3D perspective of the seismic line NOR02, after combining in transparency the EN attribute with
the PR attribute (EN+PR). The reflectors characteristics and a pattern of discontinuities are clearly visible at different levels
of detail, and the link with the faults at surface is again proposed (red segments on the top). The two boxes (blue and black
colours in Fig.5a, respectively) point out the most representative seismic facies described above. The Fig. 5b and 5c display
a comparison of the signature of reflector H in the standard amplitude image line (SA) (Fig. 5b) and, in a version including
PR attribute in transparency with SA. Again, in the inserts Figs. 5d and 5e, an analogous data comparison shows the scarce
detectability of the dense pattern of steep discontinuities in the original seismic profile. The Fig.5e displays the enhancement
obtained by plotting the PR attribute plus SA in transparency; this image greatly improves the visualization of the
fragmentation of the reflectors.
An analogous 3D multi-display of attributes EN and PR is proposed in Fig. 6a for the seismic line NOR01. The comparison
between the original line (blue box in Fig. 6b) and the EN+PR (Fig. 6c) shows the improved and peculiar signature of the
strong reflector H. The black box again reports the original plot *vs*. the PR+SA, which clearly boost the visualization of the
high-angle discontinuities.
**6 Data Interpretation: new elements and insights on the deep geological structure of the study area.**
The comparison between the original seismic data and the images obtained by the attribute analysis ensures an easier and
more detailed interpretation of the geophysical features, allowing to extend the surface geological data in depth. The
geological interpretation of these features requires a thoughtful comparison and, a calibration with other data available in the
area, e.g. geological and structural maps, co-seismic ruptures, high-resolution topography and main shocks hypocentres. The
seismic attributes provide a multiple view of the original data through the enhancement of different physical quantities.
Therefore, peculiar geophysical signatures have been detected delineating interpretative criteria (e.g. high amplitude
reflectors, phase discontinuities, fragmented reflectors patterns etc…). Such geophysical features, after a first order
interpretation, correlate well with the main outcropping geologic structures. Using the same interpretation criteria, other
surface-uncorrelated discontinuities, poorly visible in the original images (amplitude lines), are apparent at a more detailed
scale after the attribute analysis. In addition, deep reflectors showing a common signature have also been recognized,
revealing a regional character. The geological meaning and the relation of such geophysical features with the surface
geology and, with the hypocentre location of the main earthquakes are hereafter discussed.
Fig. 7 reports a global pseudo-3D view of the study region summarizing all the data analysed across the area, together with
all the faults mapped at surface (Fig. 7a) and the location of Mw 6.5 mainshock (30th October 2016). The two seismic
images in Figs. 7b and 7c have been obtained using again a multi-attributes visualization, overlapping the PR and EN
attributes in transparency plots with the original seismic lines NOR01 and NOR02, following the same procedure used for
the images in Figs. 5 and 6. The Figs. 7d and 7e propose an interpretation of the geophysical features being associated to the
faults highlighted after an accurate analysis of the discontinuities of attributes signatures, as shown in fig. s5. Regarding the
deeper parts of the sections, reflector H (blue arrows and dashed line) highlighted in NOR01 (and in CAS01), presents a
seismic character and an attribute signature compatible with the deeper reflector in NOR02 beneath CNb. This set of
reflectors is interpreted as a high acoustic impedance contrast, possibly related to an important velocity inversion occurring
between the Triassic Evaporites (anhydrites and dolostones, $Vp \approx 6$ km/s, e.g. Trippetta et al., 2010) and the underlying
acoustic Basement (metasedimentary rocks, $Vp \approx 5$ km/s, sensu Bally et al., 1986). Comparable deep and prominent
reflectors were detected also in other legacy data across adjacent regions of the Umbria-Marche Apennines (e.g. Barchi et
al., 1998; Mirabella et al., 2008). This fact confirms its regional importance, particularly because it represents a lithological
control indicating a seismicity cutoff (Chiaraluce et al., 2017; Mirabella et al., 2008; Porreca et al., 2018; Mancinelli et al.,

339   2019).

As already pointed out in the previous figures, the continuity of the deep reflector H is interrupted in the western edge by the
low-angle west-dipping discontinuity T crossing NOR02 (Fig. 7e), and not identified in the interpretation by Porreca et al.
(2018). This deep discontinuity can be interpreted as a regional thrust emerging at the footwall of the MSt, in an easternmost
sector of the region, and corresponding to the Acquasanta thrust (Centamore et al., 1993).
In NOR01, the most visible high-angle seismic discontinuity is marked by an E-dipping fault, bordering the western area of
Nb (Fig. 7d). The location and geometry of this fault, whose presence is still debated in literature, perfectly correlates with
its supposed position at surface (Blumetti et al., 1993; Pizzi et al., 2002; Galadini et al., 2018; Galli et al., 2018). Therefore,
it may represent the first clear geophysical evidence at depth of the antithetic normal fault of Norcia (aNf), suggested by
morphological studies (Blumetti et al., 1990) and paleoseismological records (Borre et al., 2003) and, belonging to a
conjugate tectonic system (Brozzetti & Lavecchia, 1994; Lavecchia et al., 1994).
The other principal structure is a synthetic (W-dipping) high-angle, normal fault bordering the eastern flank of Nb
("Nottoria-Preci fault" – Nf, Calamita et al., 1982; Blumetti et al., 1993; Calamita & Pizzi, 1994). The Nf in NOR02 is
marked by a downward propagation of a steep alignment (continuous red line in Fig. 7d). This area is also fragmented by
several minor strands parallel to the main faults (dashed lines in Fig. 7d). In particular, several west-dipping minor faults are
observed in Fig. s5a, where the shallower high-amplitude reflectors of the PR attribute are clearly disrupted.
Another discontinuity interpretable as a deep fault is visible slightly eastward, close to the mainshock hypocentral location
(Fig. 7e). This E-dipping discontinuity, emphasized by the attribute analysis, does not reach the surface. The presence of this
blind fault has been suggested by several authors in relation to the occurrence of an aftershock (Mw 5.4), which "ruptured a
buried antithetic normal fault on eastern side of Nb, parallel to the western bounding fault of CNb" (Chiaraluce et al., 2017,
Porreca et al., 2018 and Improta et al., 2019).
The central portion of NOR02, corresponding to CNb, shows a peculiar reflection fabric, dominated by high-angle
discontinuities, it is interpreted as two opposite-dipping normal faults bordering the basin, correlating with their positions
mapped at the surface (cfr. Pierantoni et al., 2013). The main fault is here represented by the W-dipping Vf, reactivated
during the 2016 earthquake (e.g. Villani et al., 2018a) which can be traced from its surface expression downward to
hypocentre location. Parallel to the Vf, several high-angle seismic discontinuities representing minor normal faults cross-cut
the gently W-dipping reflectors (Fig. 7e, further details in Fig. s5).
Analogous considerations can be extended to a multitude of E-dipping steep discontinuities at the westward side of CNb.
These may represent the evidence of an antithetic fault (aVf), and several minor fault strands characterized by high-angle dip
at shallow depths (Villani et al., 2018b). Such a fault appears connected at about 2-3 s to the W-dipping master Vf,
producing a conjugate system geometry like observed at Nb (Fig. 8e). At depth of 3.2 s, Vf fault clearly interrupts the
continuity of the top basement reflector H, whilst the relationships with the Acquasanta thrust (low-angle discontinuity T) is
more ambiguous.  Two alternative interpretations can be proposed, schematically represented in Fig. 8. In Fig. 8a, we
propose a model in which Vf merges into the deep Acquasanta thrust, suggesting a negative inversion, as a mechanism
proposed by other authors (e.g. Calamita and Pizzi, 1994; Pizzi et al., 2017; Scognamiglio et al., 2018). In Fig. 8b, Vf cuts
and displaces the Acquasanta thrust, following a steeper trajectory (ramp) (Lavecchia et al., 1994 and Porreca et al., 2018).
For both Norcia and Castelluccio di Norcia basins, the interpreted data suggest two slightly asymmetric fault systems. These
are due to conjugate sets of seismogenic master faults (Ramsay & Huber, 1987) producing a "basin-and-range" morphology
(Serva at al., 2002), progressively lowering the topography from east to west, and producing two major topographic steps,
corresponding to the CNb and Nb, respectively. Such fault systems control the evolution of the continental basins, and are
associated with several complex sets of secondary strands building up complex fault zones. Such fault strands are able to
produce surface ruptures in future earthquakes, as occurred in the 2016-2017 seismic swarm, and would require further
studies through high-resolution geophysical investigations (e.g. Bohm et al., 2011 and Villani et al. 2019).
The results of the seismic interpretation proposed in this work, supported by the attribute analysis, suggests that such
synthetic and antithetic tectonic structures at the Norcia and Castelluccio di Norcia basins cannot be actually simplified as a
unique fault plane, but they could be interpreted as complex and fractured fault zones, as conceived by Ferrario and Livio
(2018) as "distributed faulting and rupture zones".

### Conclusions

Taking into account the important role that seismic attributes play in the O&G industry, their usage might be of high interest
and impact also for improving the geological interpretation of vintage seismic data, aimed to other scientific objectives.
When applied to seismically active areas, this analysis may contribute to constrain the buried geological setting. Legacy data
powered by seismic attributes, when combined with seismological data (i.e. focal mechanisms and accurate earthquake
locations), may have high potential impact for the identification and characterization of possible seismogenic structures
(sources) and, eventually on earthquakes hazard assessment. This contribution presents one of the first case studies in which
a seismic attribute analysis is used for seismotectonic purposes, specifically on legacy seismic reflection data, in this case
collected more than 30 years ago in Central Italy. Such industrial data, nowadays irreproducible in regions where the seismic

exploration is forbidden or difficult to acquire, represent, despite the limited/poor quality, a unique source of information on the geological setting at depth.

This contribution reveals that the use of seismic attributes can improve the interpretation for the subsurface assessment and structural characterization. Certainly, the overall low quality of the data sets did neither allow to extract rock petrophysical parameters, nor more quantitative information. However, the attributes aid the seismic interpretation to better display the reflection patterns of interest and provided new and original details on a complex tectonic region in Central Italy. Our attribute analysis considerably improved the overall interpretability of the vintage seismic lines crossing the epicentral area of the 2016-2017 Norcia-Amatrice seismic sequence. In particular, we detected peculiar seismic signatures of a deep horizon of regional importance, corresponding, most probably, to the base of the seismogenic layer, and to the location and geometry of the complex active fault zones. Those consists of several secondary synthetic and antithetic splays in two Quaternary basins. These fabrics correlate with the mapped main structures at the surface. But our interpretation also reveals the existence of several faults with no clear surface outcrop, issue currently much debated in the literature. The analysis and integration of the seismic attributes allowed the determination of the deep continuation of the (known and supposed) faults and, the recently mapped co-seismic ruptures at surface, providing a pseudo-3D picture of the buried structural setting of the area. The seismic attributes may help to reduce the gap between the surface geology and deep seismological data, also revealing a high structural complexity at different scales, which cannot generally be detected only by using traditional interpretation techniques. This approach has shown the potential of the attribute analysis, that even when applied on 2D vintage seismic lines, may significantly increase the data value. For all these reasons, we strongly encourage its application for seismotectonic research, aimed to provide new information and additional constraints across seismically active regions around the world, thus contributing to hazard analysis.

**Acknowledgments**

We are grateful to Eni S.p.A. for providing an inedited set of seismic reflection lines after the 2016-2017 seismic crisis in Central Italy (raw data available in Fig.2 of supporting information). The original seismic reflection lines used in this study are available in the supplementary material, as well as the high-resolution Figures 2,3,4,7. The authors are very grateful to dgB Earth Sciences and to QGIS teams for providing the academic software used in this work. We thank Dr. Christian Berndt and Dr. David Iacopini for their valuable comments provided for this paper. We also thank the two anonymous reviewers for their patience in providing useful suggestions and detailed corrections that considerably improved this work.

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

741    ---

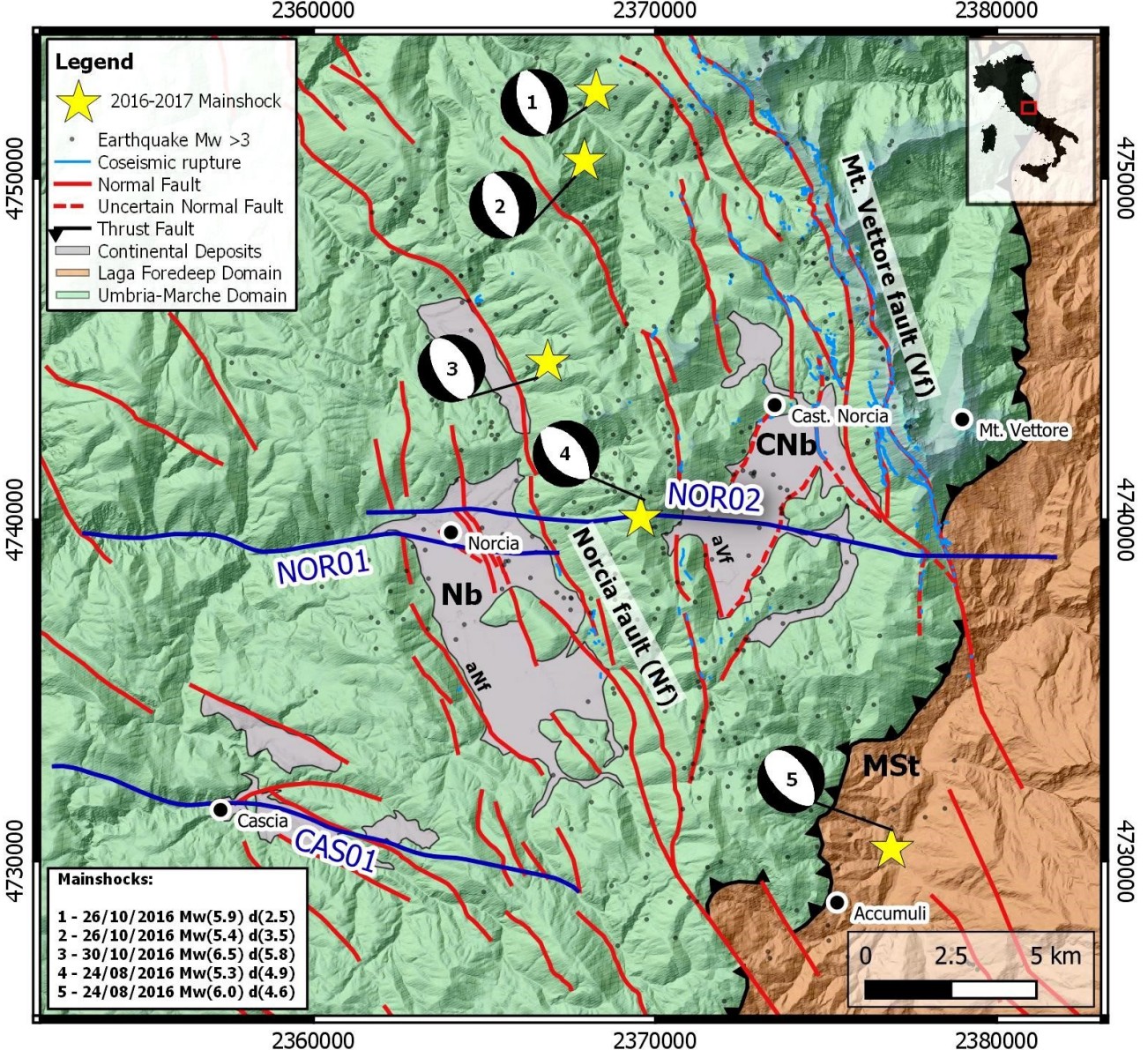

**Figure 1**

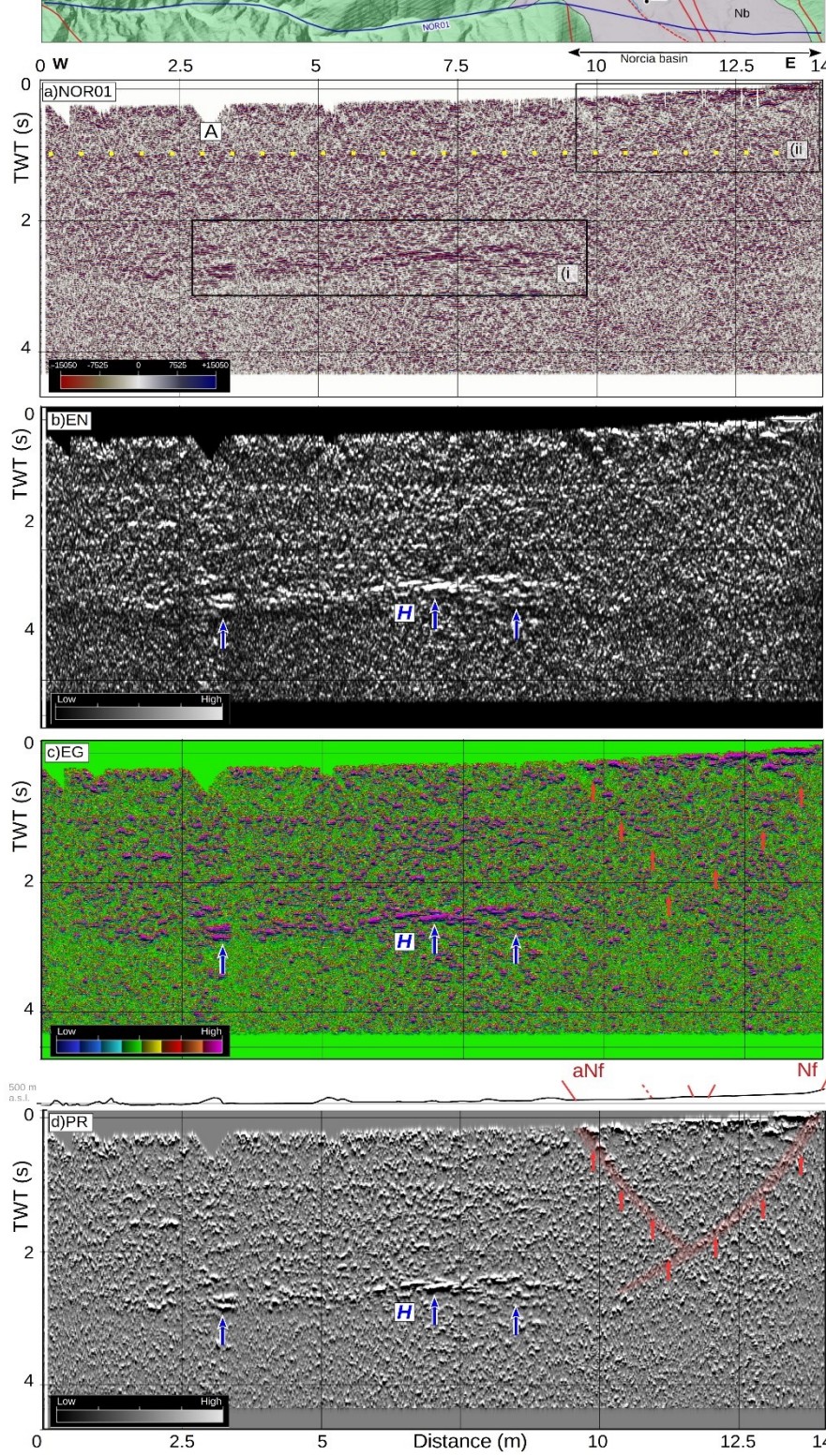

**Figure 2**

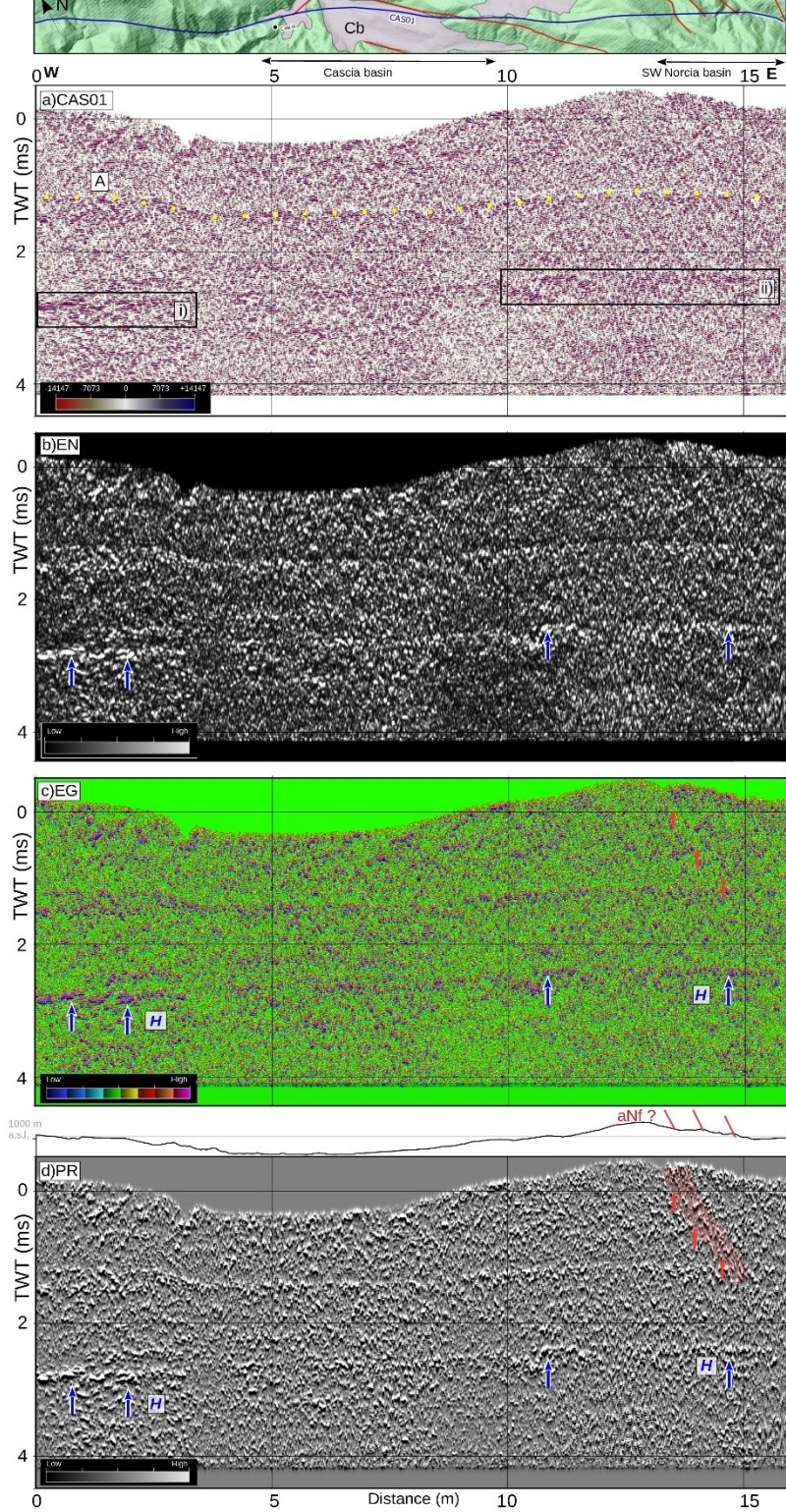


**Figure 3**

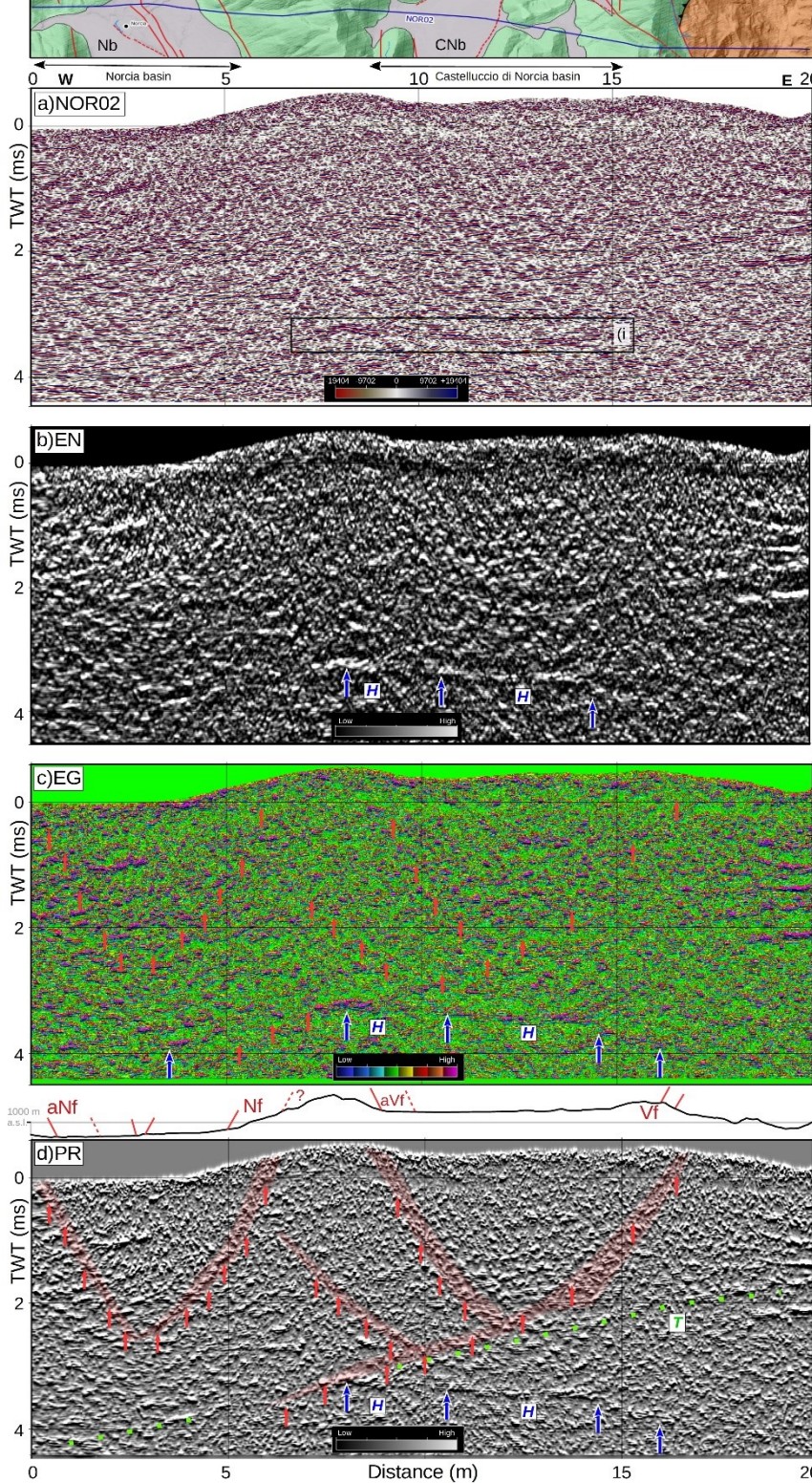


**Figure 4**

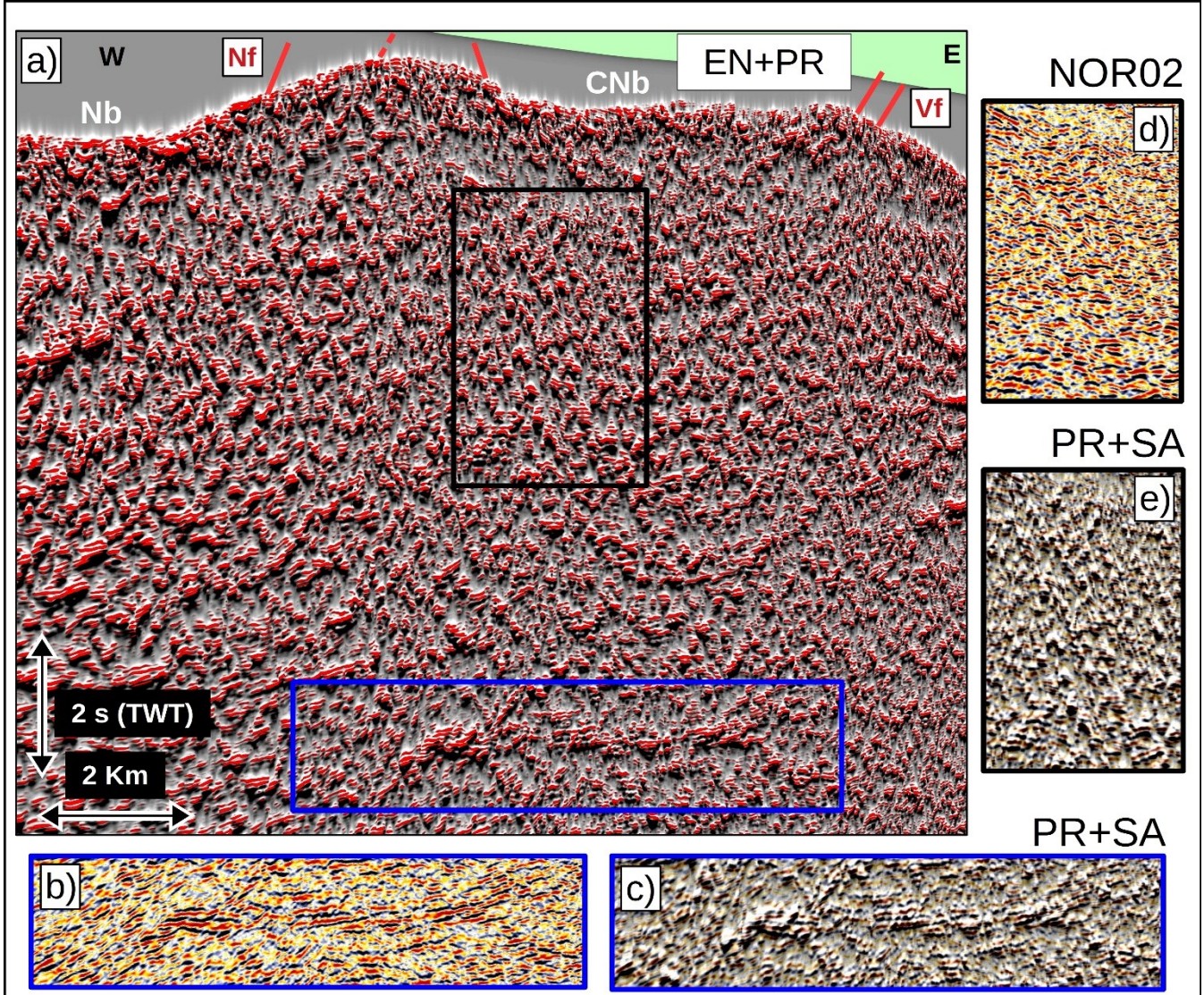

**Figure 5**


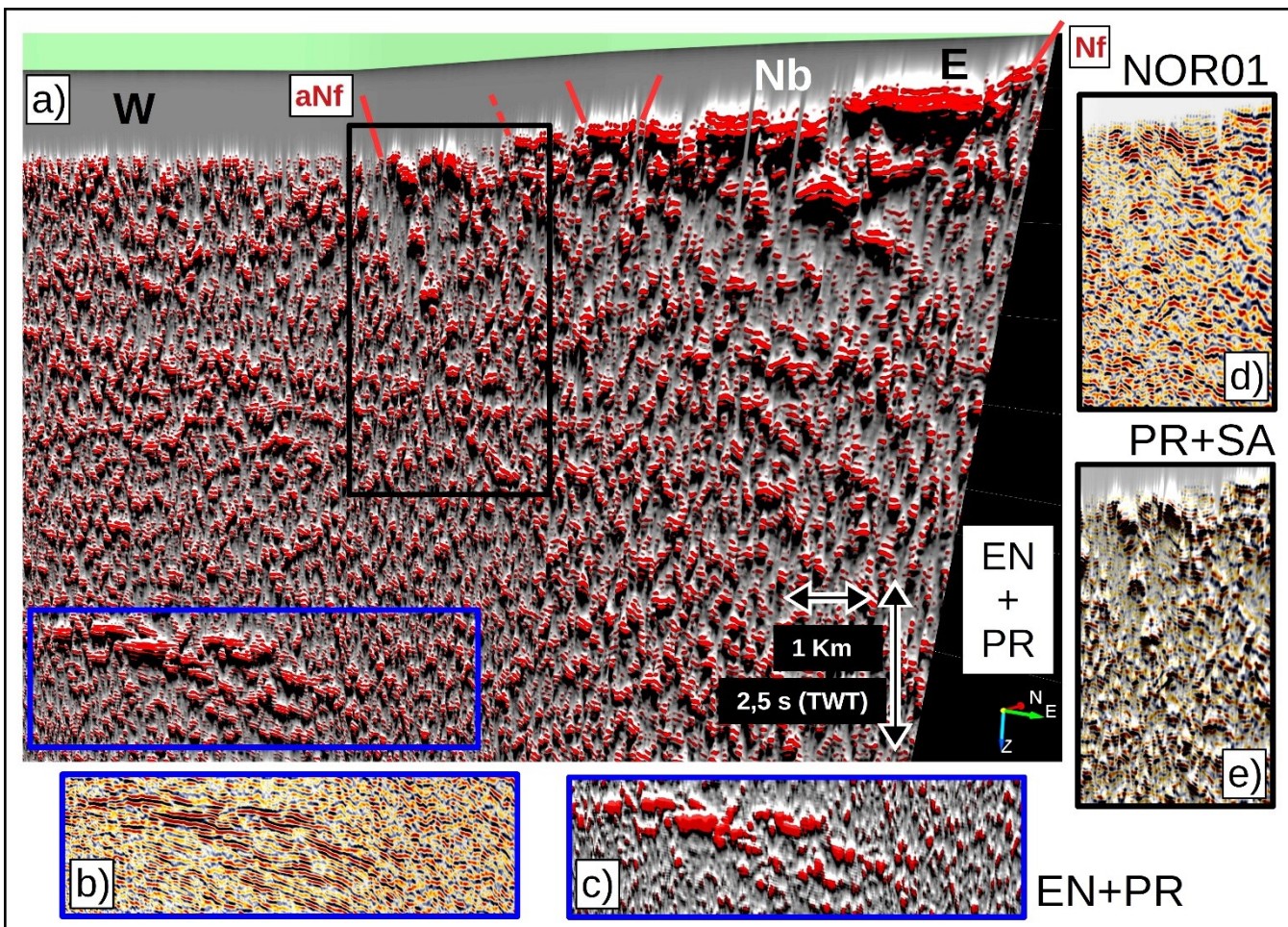

**Figure 6**

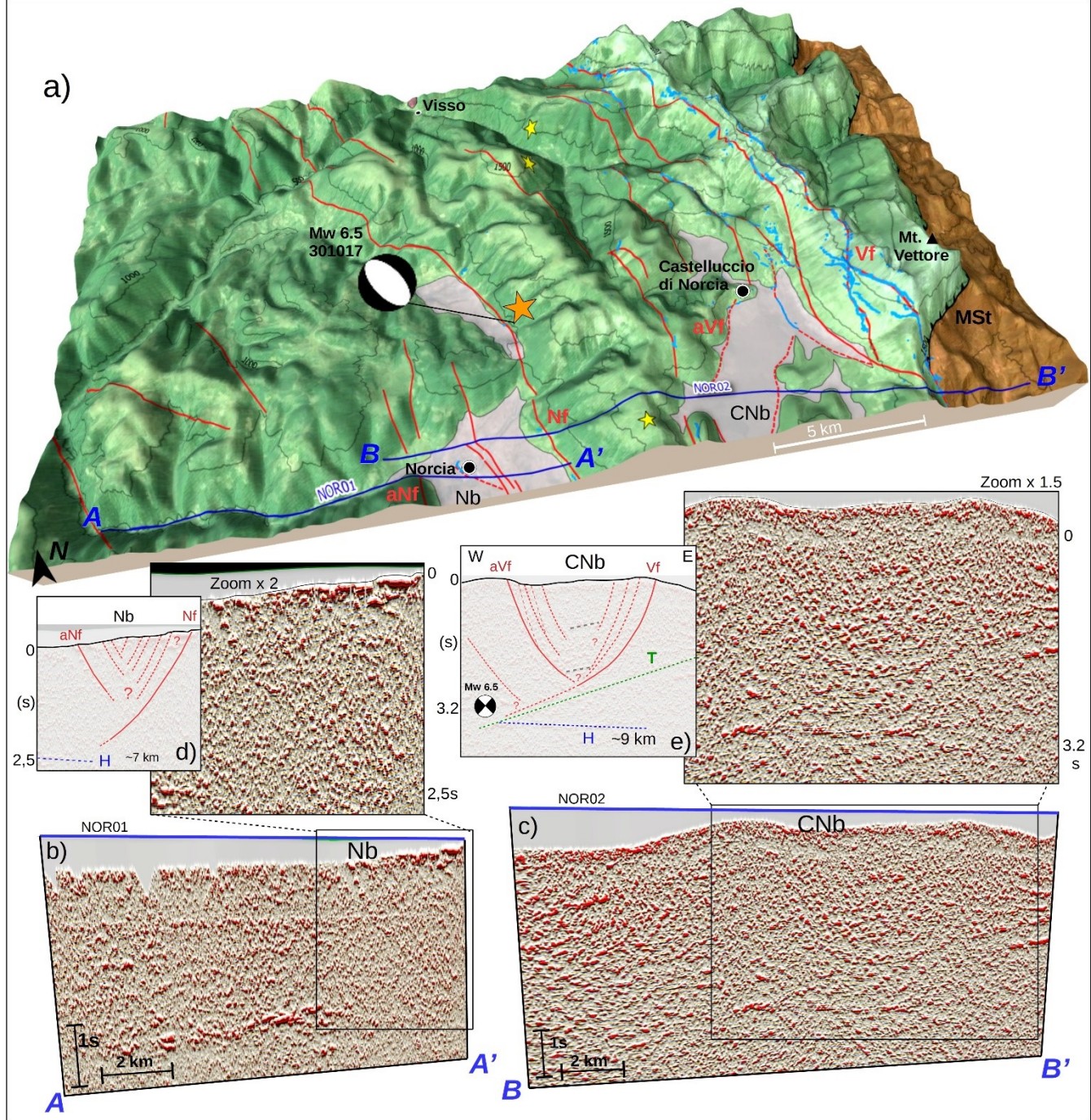


756                                                                                                 **Figure 7**


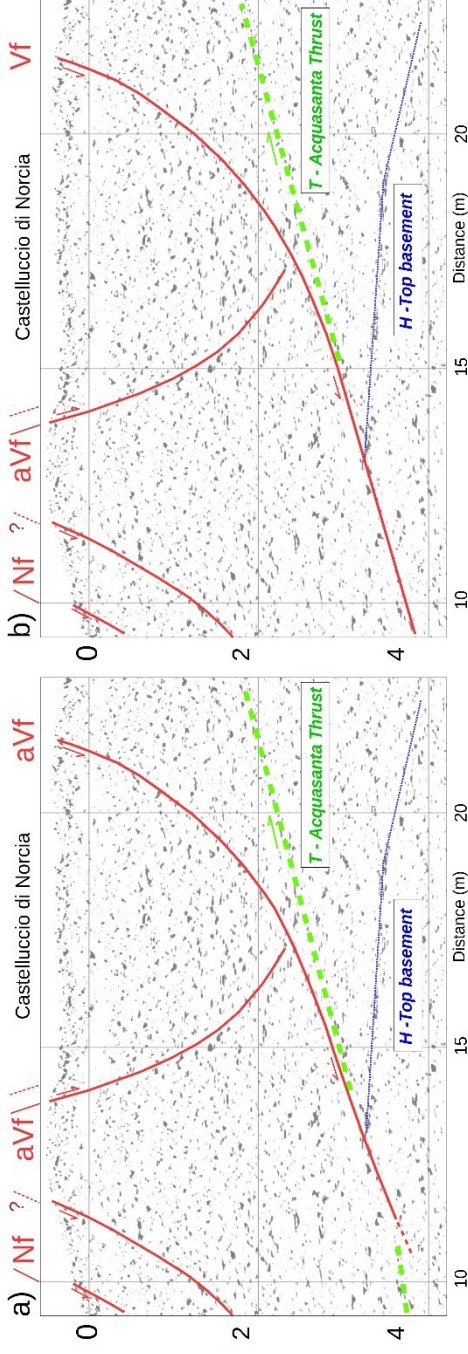


**Figure 8**
**Table 1**

| Parameters | NOR01 | NOR02 | CAS01 |
|---|---|---|---|
| **Source** | Vibroseis | Explosive | Vibroseis |
| **Length (km)** | 14 | 20 | 16 |
| **Number of traces** | 938 | 825 | 1069 |
| **Samples/trace** | 1600 | 1750 | 1600 |
| **Time window (ms)** | 6400 | 7000 | 6400 |
| **Sampling interval (ms)** | 4 | 4 | 4 |
| **Trace interval (m)** | 15 | 25 | 15 |
| **Mean Spectral amplitude (dB)** |  |  |  |



**Figures and Tables captions:**

**Figure 1: Simplified geological map of the study area (modified after Porreca et al., 2018), showing the location of the 2D seismic reflection lines. The location of the 2016-2017 mainshock are indicated by beachballs including the earthquakes magnitude. The surface ruptures and the known master faults are also highlighted. Norcia basin (Nb), Castelluccio di Norcia basin (CNb), Monti Sibillini Thrust (MSt), Mt. Vettore fault (Vf), antithetic (aVf), Norcia fault (Nf), antithetic Norcia fault (aNf).**

**Figure 2: Conventional stack image of the NOR01 transect; a) image generated by a conventional seismic reflection amplitude line (no attributes applied). Standard amplitude image refers to this conventional processing flow. The top inset depicts the main faults mapped at surface. "A" underlines a processing artefact. Boxes i) and ii) indicate the clearest reflectors; b) Energy attribute enhancing a strong reflectivity contrasts (H); c) Energy Gradient, improving the detection of dipping alignments and continuity of reflectors; d) Pseudo-Relief attribute that enhances the reflection patterns cross-cut by steep discontinuities. Nf Norcia fault, aNf antithetic Norcia fault at surface, yellow dots = A, blue arrows = H, red arrows = indication of the main lineaments and areas with major discontinuities, features are highlighted by the attributes.**

**Figure 3: Conventional stack image of CAS01: a) standard reflection amplitude image line. The top insert emphasizes the main faults mapped at surface. The label A indicates a processing artefact. Boxes i) and ii) indicate the main visible reflectors; b) Energy attribute image c) Energy Gradient attribute image; d) Pseudo-Relief image, showing the strong regional reflector H. A high-angle discontinuity on the western margin corresponds with the southern extension of aNf inferred at surface. aNf antithetic Norcia fault map at the surface, yellow dots = A, blue arrows = H, red arrows = emphasize the main lineaments and main signal discontinuities enhanced by the attribute's analysis.**

**Figure 4: Time migrated image of NOR02 profile; a) standard reflection amplitude image of the profile., The inset indicates the main faults mapped at surface; Box i) points out the most visible reflector b) Energy attribute image displaying the reflector H and a possible low angle discontinuity (T); c) Energy Gradient attribute image, showing the main lineaments detected; d) Pseudo-Relief attribute image, improving the reflectors continuity/discontinuity and the display of the areas with main signal discontinuities (red polygon) after the attribute computation. Nf Norcia fault, aNf antithetic Norcia fault; Vf Mt. Vettore fault, aVf antithetic Mt. Vettore fault at surface, yellow dots = A, blue arrows = H, green dots = T, red arrows = indication of the main lineaments**

**Figure 5: Composite multi-attribute display of NOR02, displaying the position of the main faults at surface in relation to their deep seismic attribute signature; a) Energy+Pseudo-Relief attributes, the seismic facies in the blue box is compared with the original amplitude image of the transect (b) and Energy+Pseudo-Relief (c) for comparison; the same plot for the black box is reported in figures d) and e) (original line and Pseudo-Relief+Standard Amplitude, respectively).**

**Figure 6: Composite multi-attribute rendering of NOR01, displaying the position of the main faults at surface in relation to their deep seismic attribute signature. a) Energy+Pseudo-Relief attributes, the seismic facie in the blue box shows a strong set of deep reflectors compared with the original amplitude image of the seismic profile. b) and Energy+Pseudo-Relief c). An analogous plot of the black box reported in figures d) and e) the original amplitude image of the line and the combination Pseudo-Relief+Standard Amplitude.**

**Figure 7: Integration of the surface and subsurface data; a) 3D-view (DTM by Tarquini et al., 2012) of a W-E section crossing the Norcia and Castelluccio di Norcia basins (Nb and CNb), and the main-shock locations (ISIDe working group, 2016). Surface and deep data allow to correlate the master faults and coseismic ruptures mapped at the surface. The composite multi-attribute display of NOR01 (b) and NOR02 (c), is obtained overlapping the reflection amplitude in a transparency mode with the Pseudo-Relief and Energy attributes (red palette). The black boxes centred on Nb and CNb have been magnified. An important improvement of the subsurface images provides additional details on the seismogenic fault zones: the sketches d) and e) show an interpretation reporting two conjugate basins, showing master faults along the borders and several minor synthetic and antithetic splays.**

**Figure 8:** The figure proposes two alternative interpretations of the relation between the normal Vf, the deep Acquasanta thrust (T) and, the Top- Basement reflector (H). Fig. 8a reports a model in which Vf merges into the deep Acquasanta thrust, suggesting a negative inversion, similar to the models proposed by some authors (e.g. Calamita and Pizzi, 1994; Pizzi et al., 2017 Scognamiglio et al., 2018). In Fig. 8b, Vf cuts and displaces the Acquasanta thrust, following a steeper trajectory (ramp) as proposed by other researchers (Lavecchia et al., 1994 and Porreca et al., 2018; 2020).

---

**Table 1:** List of some parameters extracted from SEG-Y headers and, the three mean frequency spectra of the three seismic lines. An approximate vertical resolution equal to 75 m has been estimated using a v=6 km/s.

---

**Fig.s1:** Figure summarizing the three original seismic reflection profiles in standard amplitude images are used in this work.

**Fig.s2:** Figure 2 reporting the computed seismic attributes without any line drawing and labels.

**Fig.s3:** Figure 3 reporting the computed seismic attributes without any line drawing and labels.

**Fig.s4:** Figure 4 reporting the computed seismic attributes without any line drawing and labels.

**Fig.s5:** The image is a magnified version of two portions of NOR01 and NOR02 profile, they are focused on the two basins of Norcia and Castelluccio di Norcia. This images aim to better display the discontinuities enhanced by the Pseudo Relief; a) PR on the Nb and, the interpretation of the primary (continuous lines) and secondary faults (dashed lines); b) PR on the CNb and interpretation of the primary (continuous lines) and secondary (dashed lines) faults bordering the basin. The continuous red lines indicate the primary normal faults bounding Nb, while the dashed red segments compose a pattern of possible secondary splays within the basin.