# Peer review of "Using Seismic Attributes in seismotectonic research: an application to the Norcia's Mw=6.5 earthquake (30th October 2016) in Central"

_Solid Earth, 2019_

## Referee Comment (RC1) · Anonymous Referee #1 · 1 Aug 2019

Ercoli et alii discuss the use of seismic attributes, applied to vintage seismic reflection data, for enhancing the structural interpretation and faults recognition with seismotectonic purposes. They present a case study by analyzing 3 vintage lines crossing the area interested by the 2016-2017 Central Italy seismic sequence. The study area is provided with updated geological maps, a dense cloud of earthquake foci and some moment tensor solutions following the 2016-2017 earthquake sequence and a dataset of earthquake-related surface ruptures, as well. This manuscript is quite well-written and the dataset worth publication, nevertheless this work needs some major revisions, due to i) a badly addressed paper scope, ii) poor quality of the graphics in their present form and iii) the somehow confusing way the data and interpretations are reported. I'm

attaching an annotated version of the manuscript with many notes and suggestions; however, the major points of concern are summarized below:

- Data and interpretations are presented in a confusing way. It is really difficult to follow the description of the recognized seismic features by means of a purely qualitative pattern recognition. Graphics are not helpful in this sense and the lack of univocal codes for e.g., faults an all the figures is making things worse. See the annotated text.

- The main point of the paper is that the use of seismic attributes can help in perform a better structural interpretation, in particular if applied to seismotectonic studies. Some seismic features are here described through a qualitative approach and a possible interpretation is proposed. If the main target of the work is to show the usefulness of the seismic attributes an external dataset is needed for validation, but this is presently lacking. The use of seismic attributes allowed to identify a possible set of secondary structures, near the surface, in both the Castelluccio and Norcia basins, and to propose the presence of an antithetic fault bordering the Norcia basin to the west. Such an interpretation is not compared to detailed geological maps (only the main structures are shown but geology is not discussed (e.g., comparing possible offset from surface geology with geophysical data). As a result, the comparison with mapped faults is only qualitative and quite poor. Moreover, the seismotectonic implications of the new interpretation is totally overlooked in the discussion and/or conclusions. In this line, I would suggest changing the title: in the present form your focusing the attention on seismotectonic research it's a really side story in the present form. A possible way to solve the lack of validation would be to make two different interpretations, with and without attributes, on the same dataset, basing interpretation on objective and declared principles (e.g., cutoff, peculiar seismic facies, direct fault detections, axial surfaces dying out… etc.) and finally compare the results with published geological maps and or sections, including the discussion on opposite interpretations in literature.

- Some recent works (see a note in the text – I'm reporting here e.g., Iacopini et al. 2016 - Iacopini, D., Butler, R. W. H., Purves, S., McArdle, N., & De Freslon, N. (2016). Exploring the seismic expression of fault zones in 3D seismic volumes. Journal of Structural Geology, 89, 54-73.) proposed the use of seismic attributes for fault recognition. One of the advantages of this and other works is that you can produce a quantitative analysis of the wavelet, filtering out, on a statistical basis, the most probable fault plane locations. This could be helpful especially in cases where a direct detection of the seismic features is problematic. Any quantitative approach is lacking in this work: at least you should discuss the attribute range and distribution in the areas where you assume the fault should be located. I would strongly suggest trying a quantitative approach, at least a descriptive one.

In summary, I had the impression that the aim of the work, as presently stated, is only partially achieved if an external dataset is not used for a detailed validation. Conversely, some interesting observations are arising from the Authors' interpretations: the presence of an antithetic fault in the Norcia basin, the deep thick-skinned thrust in NOR-2 section and the amount of possible distributed faults in the two basins. These points would benefit from more detailed discussion and comparison with present proposed models in literature. Finally, you surely have to expand the seismotectonic implications from your new interpretation. I'm sure the Authors will be able to face these criticisms and I hope that these notes will be useful to improve the present manuscript.

Please also note the supplement to this comment:
https://www.solid-earth-discuss.net/se-2019-108/se-2019-108-RC1-supplement.pdf

---

## Author Comment (AC1) · 30 Aug 2019

*Interactive short reply to comments provided by Rewiever1, se-2019-108-RC1-supplement.*

*Solid Earth Discussion Paper:*

**Using Seismic Attributes in seismotectonic research: an application to the Norcia's Mw=6.5 earthquake (30th October 2016) in Central Italy.**

Maurizio Ercoli, Emanuele Forte, Massimiliano Porreca, Ramon Carbonell, Cristina Pauselli, Giorgio Minelli, Massimiliano R. Barchi.
* * *
Dear reviewer1,

thank you for your positive comments and very useful suggestions on our manuscript.

Obviously, we have to wait for additional comments from all the reviewers before providing a complete review of the paper. However, we here anticipate that we intend to address most of your comments and suggestions, and we add some further comments on your review, which will be very useful for improving our manuscript.

We have already addressed all the minor corrections, whilst regarding the main points of concern, here below you can find our current reply:

1) We will improve the order and structure of the text regarding the data presentation, better distinguishing the data from the interpretations. We will use univocal codes for the faults and other features in the images, as suggested.

2) We are improving the main scope of this work, that aims to demonstrate how the seismotectonic studies based on seismic interpretation can benefit of the enhancement provided by attribute analysis, even in case of *vintage and poor-quality data.* **Such data are often the only available data for seismo-tectonic researches, and are worth to be use in the most appropriate way for constraining the subsurface geological setting.**

Moreover, we understood from the reviewer's comments that we have to reinforce and discuss in greater detail the seismotectonic implications of our work, highlighting the impact on the comprehension of the presented case-study.

We strongly agree with the reviewer on the need of using the external datasets to validate the proposed interpretations. So, we want to clarify how the fault distribution at the surface (as derived from analysis and comparison of ALL the existing maps) fits with the faults traced in the seismic interpretation. Such data includes a detailed summary of the main normal faults and surface ruptures of the area, obtained after carefully checking the most important geological maps in literature. We specifically refer to Pierantoni's map, but we have looked also at other regional geological maps at 1:10'000 and 1:100'000 scales, as well as the most recent ones published in literature. The other important external data-set consists of the seismological data, i.e. location and fault geometry as suggested by the focal mechanisms of the mainshocks and by the distribution of the aftershocks.

In addition, we'd like to remark that there is any deep borehole stratigraphy available for this zone (all details about surrounding deep wells have been already summarized in Porreca et al., 2018).

However, following your advice we are going to improve the geological details on the faults adding two different interpretations (with and without) on the same dataset, using clear and declared principles. In addition, we will surely reinforce the seismotectonic discussion, to better fit the manuscript title.

3-4) Thank you for suggesting additional articles about seismic attributes. Approaches like in Jacopini et al. are able to provide more quantitate results, specifically on 3D offshore data volumes that allow to test more

possibilities and have optimal spatial resolution; on 2D lines the computation of attributes has clearly more limitations, so we emphasise that the results are possibly limited, but we think this is definitely important when no other data are available.

So, we started our analysis by using first well-known and widely used attributes like the instantaneous amplitude, phase, frequency, and their combinations. Then, we have also tested attributes more suitable for 3D volumes (like coherency/similarity etc...) but they didn't perform so well due to limited vertical and spatial resolution of the data. Therefore, among the several attributes computed, we have selected the Energy, Energy Gradient and Shaded Relief, that provided, in our opinion, the best results.

Anyway, as suggested, we will try to improve the manuscript discussion and description of the attribute results on the Fault Zones.

Yours sincerely,

Maurizio Ercoli and co-authors.

---

## Short Comment (SC1) · 15 Sep 2019

Dear Editor

I have been reading with extreme interest the paper submitted by Maurizio Ercoli et al. The paper transfers a long lived but also a now very sophisticated methodology of image processing/seismic attributes, developed within the O&G exploration domain, into the regional and fault interpretation using 2D seismic lines for seismic tectonic purposes. The aims of seismic attributes in this context is to unravel and enhance deep reflectors but also high angle features and then reframe those interpretative results to fine tune the discussion Norcia Mw earthquake. I am, in fact , rather surprised it took

so long to use those techniques in this context, therefore I welcome this paper and I take the opportunity to make some constructive comments to the discussion triggered by this paper. I will be focusing more on the methodological aspect (given I have no experience on the Norcia seismotectonic area so I cannot make any comment on the tectonic implication of the results proposed). a) Frequency content: the authors introduce the properties of seismic stating that " The average frequency spectra display bandwidths ranging from few Hz up to 60-70 Hz, whilst NOR02 extends up to 100 Hz". Could they please clarify the significance of those numbers? usually a seismic line (especially onshore) barely reach those high frequency content below 1-2 second of depth. Therefore I am curious to see what the frequency decomposition and the distribution of those frequencies across the seismic line (through depth) look like. Source frequency in fact only partly relate with the frequency of the impulse signal coming from the source utilized..as the impulse will then convolve with an earth model losing by multi reflection and absorption the energy and therefore frequency content. Even a simple instantaneous frequency image would help. A frequency decomposition may help (if an interval velocity model can be assumed) to further constrain and understand the resolution, therefore estimate the thickness and therefore discuss the significance of some of the main reflectors. Given that there are no well core and well log to tie the seismic any sort of information to constrain the scale of those reflector need to be attempted.

b) Noise analysis. What is missing in the methodology and results description of this paper is a proper discussion of the noise content into the seismic and work done to isolate m understand and extract it before interpreting the seismic response using attributes. This is what in seismic interpretation we call conditioning process of the data. Every seismic lines or volume data include acquisition footprint, backscattered ground roll, migration operator aliasing, aliased shallow diffractions, multiples, and low reflectivity that falls below the ambient noise level. The expression of these noise features has negative value in mapping geology; such noise is also exacerbated by seismic attributes. So the author should discuss in depth the issues related to the seismic

which imply, getting back to the pre stack data and processing aspect or re run an image processing conditioning. There has been a lot of literature and there are software's or algorithms producing filters called edge preservation or structural oriented edge preservation which help the interpreter to smooth low and high frequency oriented and random noise around the structure of interest (once recognized. ); If they have not been tempted (comparing the image with attributes before and after the conditioning) that should be done to understand the seismic noise affecting the stacked image. Again the following paper should be taken into account in order to avoid to re invent the wheel with differently energy named attributes (I know those are the commercial name given into open source software): - Gersztenkorn, G., Marfurt, K.J., 1999. Eigenstructure-based coherence computations as an aid to 3-D structural and stratigraphic mapping. Geophysics 64, 1468e1479.

I also suggest to read on that line also the paper Pitfalls and limitations in seismic attribute interpretation of tectonic features Kurt J. Marfurt1 and Tiago M. Alves published into the seg AAPG interpretation: - Marfurt, K.J., Alves, T.M., 2015. Pitfalls and limitations in seismic attribute interpretation of tectonic features. Interpretation 3, 5e15. http://dx.doi.org/10.1190/INT-2014-0122.1. c) i notice that the authors have avoided to use coherency and dip related attributes . In some case they may help to unravel subtle details and more importantly to distinguish noise surrounding certain dipping structure. In some other they may be totally useless (if too much noise distributed is affecting the seismic). Again, a mention should be given by the authors if those attributes have been attempted. The papers that tempted this approach in 3D volume (which imply using modified algorithms) should be take into account when discussing the results. Those methodologies are in fact now moving beyond into detailing damage structures surrounding large scale faults , exploring strain/fault facies using various statistical and soon machine learning approach. Here some of the pioneering examples:

- C. Townsend, I.R. Firth, R. Westerman, L. Kirkevollen, M. Harde, T. Andersen Small seismic-scale fault identification and mapping Geol. Soc. Lond. Spec. Publ., 147

(1998), pp. 1-25 - Dutzer, H. Basford, S. Purves. Investigating fault-sealing potential through fault relative seismic volume analysis. Petroleum Geology Conference Series, vol. 7 (2010), pp. 509-515 - Chopra, S., Misra, S., Marfurt, K., 2011. Coherence and curvature attributes on pre-conditioned seismic dataset. Lead. Edge 32, 260e266. - Iacopini, D., Butler, R.W.H., Purves, S., 2012. Seismic imaging of thrust faults and structural damage: a visualization workflow for deepwater thrust belts. First Break 30, 39e46.I, - Iacopini et al.Exploring the seismic expression of fault zones in 3D seismic volumes, JSG 2016 http://dx.doi.org/10.1016/j.jsg.2016.05.005 - Cunningham, Jennifer Elizabeth; Cardozo, Nestor; Townsend, Christopher; Iacopini, David; Wærum, Gard Ole (2019). Fault deformation, seismic amplitude and unsupervised fault facies analysis: Snøhvit Field, Barents Sea. Journal of Structural Geology. ISSN 0191-8141. Volume 118. p. 165-180. DOI: 10.1016/j.jsg.2018.10.010. where all the attributes are carefully discussed

Another attribute who may certainly help to visualize any sort of oriented structure without adding smoothing is the instantaneous phase and/or the cosine of it ( called cosine of the phase). I suggest the following paper as reference as a nice explanation of the physics effect can be read: - Purves, S., 2014. Phase and Hilbert transform. Lead. Edge 34, 1246e1253

d) A different approach , that is now very important to guide the interpretation of certain seismic signal , come from the series of paper of the Bergen-Stavanger school running forward seismic modelling test. I suggest to read those papers and use them in the discussion when interpreting seismic, as they may be inspiring in discussing what the interpretation and acquisition pitfall who may biases the fault interpretation but also to compare what the results obtained in a more wide and up to date scientific framework.

- C. Botter, N. Cardozo, S. Hardy, I. Lecomte, A. Escalona. From mechanical modeling to seismic imaging of faults: a synthetic workflow to study the impact of faults on seismic. Mar. Petrol. Geol., 57 (2014), pp. 187-207 - C. Botter, N. Cardozo, I. Lecomte, A. Rotevatn, G. Paton. The impact of faults and fluid flow on seismic images of a relay

ramp over production time. Petrol. Geosci., 23 (2017), pp. 17-28

e) processing strategy: another approach has been taking by the Bruno&Improta work on the processing procedure to better image shallow structure using exploration data. Those need to be included in the discussion of the results obtained as well.

- Bruno et al. Ultrashallow seismic imaging of the causative fault of the 1980, M6.9, southern Italy earthquake by pre‐stack depth migration of dense wide‐aperture data . GEOPHYSICAL RESEARCH LETTERS, VOL. 37, L19302, doi:10.1029/2010GL044721, 2010 - Improta et al. Detecting young, slow‐slipping active faults by geologic and multidisciplinary high‐resolution geophysical investigations:A case study from the Apennine seismic belt, Italy. JOURNAL OF GEOPHYSICAL RESEARCH, VOL. 115, B11307, doi:10.1029/2010JB000871, 2010

I hope those comments may help to fine tune the paper and the discussion of the interpreted data proposed.

Best wishes David Iacopini

---

## Author Comment (AC2) · 20 Oct 2019

Jacopini:

Dear Editor,

I have been reading with extreme interest the paper submitted by Maurizio Ercoli et al. The paper transfers a long lived but also a now very sophisticated methodology of image processing/seismic attributes, developed within the O&G exploration domain, into the regional and fault interpretation using 2D seismic lines for seismic tectonic purposes. The aims of seismic attributes in this context is to unravel and enhance deep reflectors but also high angle features and then reframe those interpretative results to fine tune the discussion Norcia Mw earthquake. I am, in fact, rather surprised it took so long to use those techniques in this context, therefore I welcome this paper and I take the opportunity to make some constructive comments to the discussion triggered by this paper.

Authors:

We really thank David Jacopini for this positive note and for all the constructive and relevant comments. We are glad that he remarks the spirit of this work, highlighting the novelty of our study in suggesting the use of seismic attributes in seismotectonic research.

Jacopini:

I will be focusing more on the methodological aspect (given I have no experience on the Norcia seismotectonic area so I cannot make any comment on the tectonic implication of the results proposed).

a) Frequency content: the authors introduce the properties of seismic stating that "The average frequency spectra display bandwidths ranging from few Hz up to 60-70 Hz, whilst NOR02 extends up to 100 Hz".  Could they please clarify the significance of those numbers? usually a seismic line (especially onshore) barely reach those high frequency content below 1-2 second of depth.

Authors:

We meant to describe the whole frequency range of the seismic lines and in tab.1 we display the frequency contents using amplitude/frequency spectra computed on the entire time window of each processed line. In the case of NOR02 line the spectrum shows a slighter high frequency contents in the range 40-80 Hz, but basically the bandwidth is in the range 10-50 MHz, therefore we'll modify the sentence, thank you for this note.

Jacopini:

Therefore, I am curious to see what the frequency decomposition and the distribution of those frequencies across the seismic line (through depth) look like. Source frequency in fact only partly relate with the frequency of the impulse signal coming from the source utilized..as the impulse will then convolve with an earth model losing by multi reflection and absorption the energy and therefore frequency content. Even a simple instantaneous frequency image would help.

Authors:

Two images of the instantaneous frequency are provided as requested. The sections are clearly contaminated by high frequency noise, however also higher frequency components of the signal are visible in the shallow portion (0-2 seconds), whilst progressively lower frequencies are visible more in depth.

In addition to your comment, we can state that other more sophisticated and quantitative techniques, like spectral decomposition, are not useful on the data here analysed due to the overall low signal-to-noise ratio and high phase variability.

[Figure]

Distance [m] vs TWT [ms]

Distance [m] vs TWT [ms]

Jacopini:

A frequency decomposition may help (if an interval velocity model can be assumed) to further constrain and understand the resolution, therefore estimate the thickness and therefore discuss the significance of some of the main reflectors. Given that there are no well core and well log to tie the seismic any sort of information to constrain the scale of those reflector need to be attempted.

Authors:

We already reported in this work an estimation of the average resolution in the caption of table 1. We had computed for these lines a value ranging between 70-80 m assuming an average velocity of 6000 m/s and the worse scenario considering a frequency of 20 Hz. This value is sufficient to resolve the main regional reflectors belonging to the multilayer formations of the Umbria-Marche succession. Unfortunately, in addition to the low S/N, also the phase continuity is in general quite poor thus preventing a detailed analysis on, for instance, the

occurrence of interference phenomena investigated through sophisticated spectral decomposition techniques based on STFT or wavelet analysis.

Jacopini:

b) Noise analysis. What is missing in the methodology and results description of this paper is a proper discussion of the noise content into the seismic and work done to isolate m understand and extract it before interpreting the seismic response using attributes. This is what in seismic interpretation we call conditioning process of the data. Every seismic lines or volume data include acquisition footprint, backscattered ground-roll, migration operator aliasing, aliased shallow diffractions, multiples, and low reflectivity that falls below the ambient noise level. The expression of these noise features has negative value in mapping geology; such noise is also exacerbated by seismic attributes. So, the author should discuss in depth the issues related to the seismic which imply, getting back to the pre stack data and processing aspect or re run an image processing conditioning. There has been a lot of literature and there are software's or algorithms producing filters called edge preservation or structural oriented edge preservation which help the interpreter to smooth low and high frequency oriented and random noise around the structure of interest (once recognized. ); If they have not been tempted (comparing the image with attributes before and after the conditioning) that should be done to understand the seismic noise affecting the stacked image.

Authors:

The question is relevant and we agree about the importance of a detailed noise analysis and data conditioning. These seismic lines are mainly characterized by random high-frequency noise. We have made an extensive analysis and filtering tests to attenuate such components, using the steering algorithms implemented in OpendTect (e.g. 1 - Phase Gradient, FFT and PCA, also specifying 70° as maximum dip angle of the events to exclude possible (sub) vertical artefacts/noise components, and using a median filter tested using different n° traces/samples). Here below two images (Energy Gradient attribute NOR01) for comparison: without and with data conditioning with a dip-steering filter. Slight benefits can be appreciated on the continuity of reflectors as well as, for example, on continuity of the high deep antithetic fault of Norcia (the small red dots suggest its approximate surface location). However, after the data conditioning we did not observe dramatic improvements in our 2D lines, probably because the data conditioning performs more efficiently of 3D seismic volumes. For this reason, we have decided to summarized very shortly this procedure in the text, avoiding a too detailed and technical treatment to weight down the methodological section. However, we can surely introduce the problem briefly describing the data pre-conditioning, if necessary. Thank you for the suggestion and in particular for the useful references provided.

No data conditioning

[Figure]

Data conditioning

[Figure]

Jacopini:

Again, the following paper should be taken into account in order to avoid to reinvent the wheel with differently energy named attributes (I know those are the commercial name given into open source software): - Gersztenkorn, G., Marfurt, K.J., 1999. Eigenstructure-based coherence computations as an aid to 3-D structural and stratigraphic mapping. Geophysics 64, 1468e1479. I also suggest to read on that line also the paper Pitfalls and limitations in seismic attribute interpretation of tectonic features Kurt J. Marfurt1 and Tiago M. Alves published into the seg AAPG interpretation: - Marfurt, K.J., Alves, T.M., 2015. Pitfalls and limitations in seismic attribute interpretation of tectonic features. Interpretation 3, 5e15. http://dx.doi.org/10.1190/INT-2014-0122.1.

Authors:

We agree with this comment, currently there are many papers reporting the name "Energy" for this attribute, but we surely improve the attribute description in the methodological part adding these references. Regarding the papers related to the Pitfalls, we have shorty reported some references about pitfalls in the introduction (line 79), when describing the pros and cons of the seismic attribute use in two 2D seismic data. However, we'll improve the text following your suggestion.

Jacopini:

c) I notice that the authors have avoided to use coherency and dip related attributes. In some case they may help to unravel subtle details and more importantly to distinguish noise surrounding certain dipping structure. In some other they may be totally useless (if too much noise distributed is affecting the seismic). Again, a mention should be given by the authors if those attributes have been attempted. The papers that tempted this approach in 3D volume (which imply using modified algorithms) should be take into account when discussing the results.  Those methodologies are in fact now moving beyond into detailing damage structures surrounding large scale faults, exploring strain/fault facies using various statistical and soon machine learning approach. Here some of the pioneering examples: … (Literature list) Another attribute who may certainly help to visualize any sort of oriented structure without adding smoothing is the instantaneous phase and/or the cosine of it (called cosine of the phase).

Authors:

Thank you for the request. Among the tested seismic attributes we have also calculated coherency-based ones, also considering the effects of the dip on them. However, we have decided to discard them in this work, because in our opinion they are not performing particularly well on our data. As suggested, probably the noise limits their efficiency (also in the conditioned lines) and, again we speculate that they may perform better on 3D seismic volume instead of this type of 2D vintage lines. We report an example of a similarity attribute, as an example of the test performed. We'll add some references as you suggested to make the discussion more exhaustive, also suggesting possible further innovative approaches like machine learning applications on this topic. Regarding the phase attributes, we have also used these during our tests, obtaining quite helpful results. We provide an image of the cos-phase here below that highlights the lateral continuity of some interesting reflectors, representing quaternary deposits infilling the Norcia Basin (about 12.5 km along the line). In any case in the paper we have preferred the Pseudo-Relief attribute that efficiently shows the continuity/discontinuity of the reflectors linked with reflection amplitude information.

COS-PHASE applied on the conditioned seismic line NOR01:

[Figure]

Similarity attribute computer on line NOR01:

[Figure]

Jacopini:

A different approach, that is now very important to guide the interpretation of certain seismic signal, come from the series of paper of the Bergen-Stavanger school running forward seismic modelling test. I suggest to read those papers and use them in the discussion when interpreting seismic, as they may be inspiring in discussing what the interpretation and acquisition pitfall who may biases the fault interpretation but also to compare what the results obtained in a more wide and up to date scientific framework.

- C. Botter, N. Cardozo, S. Hardy, I. Lecomte, A. Escalona. From mechanical modelling to seismic imaging of faults: a synthetic workflow to study the impact of faults on seismic. Mar. Petrol. Geol., 57 (2014), pp. 187-207
- C. Botter, N. Cardozo, I. Lecomte, A. Rotevatn, G. Paton. The impact of faults and fluid flow on seismic images of a relay ramp over production time. Petrol. Geosci., 23 (2017), pp. 17-28.

Authors:

Thank you, we'll surely look at such papers and possibly improve the discussion inserting this relevant topic, which is ancillary to the main focus of our paper.

Jacopini:

e) processing strategy: another approach has been taking by the Bruno&Improta work on the processing procedure to better image shallow structure using exploration data. Those need to be included in the discussion of the results obtained as well… (Bruno et al., 2010 and Improta et al., 2010).

Authors:

Thank you again for this advice. As we have reported in the introduction, two main strategies can be attempted in this type of studies. The first is the use of an attribute analysis as we propose here; a second possibility is the reprocessing of the original shot gathers, using modern processing tools and an improved computational power. However, we had only stack and migrated seismic lines available for this specific work (as is quite common when dealing with vintage data). Only recently we have received some  original pre-stack raw data and we are currently working on a dedicated reprocessing workflow, particularly focused on the refinement of the static corrections, improved velocity analysis and interpretive processing. However, this second approach and workflow will be presented in another dedicated paper.

Jacopini:

I hope those comments may help to fine tune the paper and the discussion of the interpreted data proposed.

Best wishes David Iacopini

Authors:

We have really appreciated all your relevant comments and we are sure that they contribute to considerably improve our work.

---

## Referee Comment (RC2) · Anonymous Referee #2 · 25 Oct 2019

General comments on Ercoli et al. Submitted to Solid Earth journal

The manuscript "Using Seismic Attributes in seismotectonic research: an application to the Norcia's Mw=6.5 earthquake (30th October 2016) in Central Italy" by Maurizio Ercoli et al. submitted to Solid Earth proposes the use of seismic attribute analysis approach on three vintage reflection seismic profiles acquires across the Norcian and Castellucio di Norcia basins to determine the extension and geometry of the geological structures. This region was the epicentral area of the 2016-2017 seismic crisis in central Italy.

This manuscript could be of interest to geologists and geophysicists working in active

tectonics and using reflection seismic data. However, in my opinion, it needs still some work in the structure of the writing and, most important, more work in the interpretation of the data or, at least, it needs to show more clearly all the interpretations the authors are doing. I am not an expert in the analysis of this type of data (onshore seismic data across rocky regions) but I have many difficulties to identify the same structures the authors are interpreting. At the end, I have had the impression that the authors have extended the surface map structures in depth following some possible alignments. My question is, would have they interpreted the same structures without the surface information? To me, there is a high uncertainty in the interpretation of the alignments in the seismic profiles that, then, I have problems to believe the final structural model proposed in the manuscript.

Following there are some general comments on the different sections. I also provide a commented manuscript that hope will help to improve the quality of the manuscript and the presented results. Despite my criticism, to be intended solely as constructive, I warmly encourage the authors to make any effort for the publication of this manuscript, because of the relevance of the proposed approach and objectives.

1.Introduction

I think that in general the introduction needs to be restructured to emphasize the main aspects of what authors wants to expose. It is a very confusing introduction. I am not a native English speaker and I have found some errors, so I think that a native English speaker should review the final version of the manuscript.

Some specific comments:

Paragraph from lines 69 to 104 is a long paragraph that jumps from one idea to another and then back on. It is confusing and needs to be rewritten. Why mention 2D data vs 3D data various times? Just need to stress the differences and then stress the information and advantages of using 2D dataset, mainly which it is available and ready to work on. In addition, sentences like the one in lines 82-84 are out of sense in that paragraph.

The stated between lines 85 and 98 is confusing. This may be rewritten, but also I think that it make no sense to explain all this in the introduction.

2.Geological framework

This section of the manuscript is a little bit confusing and difficult to follow. The authors jump from one topic to another in some paragraphs and is difficult to understand the geological structure of the area. I think it is necessary some organization. Begin for the big geological units, as done. Then, explain the structures, the fault systems in the area. Continue with the basins object of study. Finally talk about the seismicity in the area and the recent earthquakes and the faults that show surface rupture. In addition, I recommend the authors to be consistent with the names of the units, faults, for example, the Laga foredeep domain is referred in three or four different ways, and that is confusing.

3.Data

The authors mention a couple of times the supporting information, but in fact the information is provided in tables and figures in the manuscript.

Also the figures in the supporting information are not correctly identified and some errors of profiles identifications are present and must be corrected.

4.Methods

Authors comments that they have tested several post-stack attributes, but it is not clear at all why they select ones and not others. Maybe it is not necessary to explain this? I am not an expert in seismic attribute analysis.

5.Results

To me it is necessary to include in the supplementary information the profiles (original and attribute analysis) without any interpretation and each one on one page at a bigger scale. The profiles on the manuscript show arrows pointing to specific features that attract the attention towards the author's interpretation. For example, in Fig2c the authors points with red arrows to some discontinuity (?) but at the same time the arrows mask reflectors around. I could point to similar features (orange arrow in the corresponding figure on my commented manuscript) that could point to a normal fault dipping to the W? That suggests me that the authors are just looking for structures that have been recognized at surface and not for all the other possible structures in the area/profiles. But again, without the un-interpreted profiles it is difficult to compare observations.

I would recommend to describe each profile independently pointing to the observations done in each attribute profile and follow the same structure from one profile to the other. Begin with the seismic section and describe what you see and what is or could correspond the observed artefacts, then, the EN section with the specific observations, after, the EG section and, finally, the PR section. This makes things easy to the reader and not necessary to jump from one profile to the other and return. I suggest to identify the different high-dipping lineaments in the figures with letters (e.g., L1, L2,...) and then refer to them in the text. It would be much easier for the reader to understand to which lineament the authors are referring.

In profile NOR02 the relationship between horizons T, H and the west-dipping lineament interpreted as bounding the CNb is not clear. In lines 256-259 it is said that horizon H is interrupted by horizon T, which crosses all the profile from east to west and dipping to the west. Later on, in lines 275-276 it is said that a west-dipping lineament truncates and disrupts horizons (discontinuities) T and H. In general to me is very difficult to interpret the lineaments in all the profiles (as pointed in a number of comments in the manuscript) but in that case I think that the authors are proposing different interpretations for the same observations. This needs to be clarified.

6.Discussion and conclusions

As said in various comments I have problems to interpret the steep discontinuities on the different seismic profiles (amplitude and attributes). All the discussion is based on the authors interpretation and since I cannot interpret the same things I cannot support it. But, I am not a specialist in this type of seismic interpretations.

Please also note the supplement to this comment:
https://www.solid-earth-discuss.net/se-2019-108/se-2019-108-RC2-supplement.pdf

———————————————————

[Figure]

**Supplement:**

[revised manuscript text omitted]

**Figures and Tables captions:**

**Figure 1: Simplified geological map of the study area (modified after Porreca et al., 2018), showing the 2D seismic data tracks, the**
**2016-2017 mainshock locations, beachballs and magnitudes, the surface ruptures and the known master faults. Nb Norcia basin,**
**CNb Castelluccio di Norcia basin.**

**Figure 2**: ck version of NOR01; a) reflection amplitude, yellow dots underline a processing artefact (A); b) Energy attribute
**enhancing a strong reflectivity contrasts (H, blue arrows); c) Energy Gradient, improving the detection of dipping alignments and**
**continuity of reflectors; d) Pseudo-Relief enhancing the reflection patterns cross-cut by steep discontinuities (red arrows). Nf Norcia**
**fault, aNf antithetic Norcia fault.**

**Figure 3: Stack version of CAS01, with same attributes computat**: a) reflection amplitude (yellow dots display processing
**artefacts); b) Energy attribute c) Energy Gradient attribute; d) Pseudo-Relief, showing the strong regional reflector H (blue arrows).**
**A high-angle discontinuity on the western margin is interpretable as a normal fault, showing an attribute signature analogous to**
**aNf.**

**Figure 4: Time migrated version of NOR02; a) reflection amplitude; b) Energy attribute displaying the reflector H (blue arrows)**
**and a possible low angle discontinuity (T, green dots); c) Energy Gradient attribute, showing the master faults bounding the basins**
**(red arrows); d) Pseudo-Relief, improving the reflectors continuity/discontinuity and the master faults display (red arrows). Nf**
**Norcia fault, aNf antithetic Norcia fault; Vf Mt. Vettore fault, aVf antithetic Mt. Vettore fault.**

**Figure 5: Multi-attribute display of NOR02; a) EN+PR attributes, the seismic facie in the blue box is compared with the original**
**seismic line (b) and EN+PR (c) for comparison; the same plot for the black box is reported in figures d) and e) (original line and**
**PR+SA, respectively).**

[Figure]

[Figure]

**Figure 6: Multi-attribute display of NOR01; a) EN+PR attributes, the seismic facie in the blue box showing a strong set of deep**
**reflectors is compared with the original seismic line in b) and EN+PR c). An analogous plot of the black box reports in figures d)**
**and e) the original line and the combination PR+SA.**

**Figure 7: Integration of surface and subsurface data (DTM by Tarquini et al., 2012); a) 3D-view of a W-E section crossing Nb and**
**CNb, and the mainshock locations (ISIDe working group, 2016). Surface and deep data allow to correlate the master faults and**
**coseismic ruptures at the surface. The multi-attribute display of NOR01 (b) and NOR02 (c), is obtained overlapping the reflection**
**amplitude in transparency with the Pseudo-Relief and Energy attributes (red palette). A significative improvement of the subsurface**
**images provides unprecedent details on the seismogenic fault zones: the two conjugate basins show master faults along the borders**
**and some minor synthetic and antithetic splays (see d) and e) sketches).**

**Table 1: List of some parameters extracted from SEG-Y headers and three mean frequency spectra of the three seismic lines. An**
**approximate vertical resolution equal to 80 m was derived (v=6 km/s).**

---

## Author Comment (AC3) · 22 Nov 2019

Dear reviewer2,

on behalf of all my co-authors I really thank you for the time spent to revise our manuscript. We are sure that all the suggestions and positive comments provided in the document "se-2019-108-RC2" will contribute to improve our work. We have also appreciated the detailed comments and corrections provided in the document "se-2019-108-RC2-supplement", that will be extremely useful during the revision.

Before providing a complete review of the paper, in this short reply we attempt to an-

swer to some of your considerations provided in "se-2019-108-RC2".

We are glad that this manuscript based on an attribute analysis application in seismo-tectonics may be of interest of geologists and geophysicists working in active tectonics. Regarding your main paper issues, we reply hereafter point by point.

1) REV2: it needs still some work in the structure of the writing and, most important, more work in the interpretation of the data or, at least, it needs to show more clearly all the interpretations the authors are doing.

Authors: As also remarked in the reply to REV1, we are surely going to improve the structure of the writing, using all your suggestions (see following points). We will also work on the data interpretation and presentation, trying to better clarifying to what extent it is constrained by the seismic data-set.

2) REV2: I am not an expert in the analysis of this type of data (onshore seismic data across rocky regions), but I have many difficulties to identify the same structures the authors are interpreting.

Authors: we explicitly declared that the quality of our data-set is limited, however, as we mentioned: a) this is very common in on-shore, rocky areas; b) they are the only available seismic data, and unravelling the subsurface structure is crucial for a better comprehension of the seimo-tectonic work. As far as possible, we will try to improve the image quality and readability of the described seismic sections, aiming to provide clear evidences also to readers with limited confidence with attribute analysis. However, we remark that seismic interpretation always encompasses some subjectivity, especially when the data are not "self-explaining", as when working with vintage datasets on mountainous areas. The use of integrated attributes can contribute to give additional suggestions and indications, that not always are self-explanatory evidences.

3) REV2: At the end, I have had the impression that the authors have extended the surface map structures in depth following some possible alignments. My question is,

would have they interpreted the same structures without the surface information? To me, there is a high uncertainty in the interpretation of the alignments in the seismic profiles that, then, I have problems to believe the final structural model proposed in the manuscript.

Authors: The answer is yes, we actually have tried, as a first instance, to extend in depth the surface structures reported in literature and mapped after the most recent geological/structural surveys, and later following possible seismic alignments enhanced by attributes. So, we have basically started by the available information, because we think that starting from the surface geological/structural data, specifically in an area characterized by overall poor subsurface records, it would be the best approach to better constrain any analysis. However, because not all the fault segments show outstanding surface expressions, we have interpreted similar alignments given by reflectors discontinuity (e.g. amplitude, phase differences highlighted by attributes) as secondary faults splays/fault zones. This research of a trade-off between surface and deep evidences of faulting is due, as remarked in the introduction, to the poor data availability for the area. Regarding the interpretation, we clearly agree that there is a certain degree of uncertainty; but we used a conservative approach, just marking the structures that are clearly "seismically" evident, that in our opinion would be also traced without surface evidences. The antithetic (SW-dipping fault) of Norcia, for example, speculated in the past by some authors without clear evidences, is imaged in our data and it is the first time that is it recognized in a reflection dataset. Finally, we are conscious that there are several different models proposed for the area: we provide new information which can be helpful in order to validate/constraint one of them. As also structural analysis of exposed outcrops, seismic interpretation has clearly a certain degree of subjectivity: for example, no one "saw" low-angle normal faults before the concept was introduced. So, in this case our interpretation is driven by the background information that: a) the studied region is affected by intense extensional tectonics, generally disrupting previous compressional belt; b) the knowledge of the position of some major normal faults exposed at the surface (possibly connected to the earthquakes mainshocks). In addition, we are totally conscious that our data-set is not comparable with modern 3-D seismic surveys of recent basins and offshore areas, where a much larger degree of objectivity can be reached by seismic interpretation.

REV2: Following there are some general comments on the different sections. I also provide a commented manuscript that hope will help to improve the quality of the manuscript and the presented results. Despite my criticism, to be intended solely as constructive, I warmly encourage the authors to make any effort for the publication of this manuscript, because of the relevance of the proposed approach and objectives.

Authors: We really appreciate the positive comments, and we'll do our best to improve the results and the overall quality of the manuscript.

REV2: 1. Introduction - I think that in general the introduction needs to be restructured to emphasize the main aspects of what authors wants to expose. It is a very confusing introduction. I am not a native English speaker and I have found some errors, so I think that a native English speaker should review the final version of the manuscript.

Authors: Thank you, we will revise the errors also using your punctual corrections and specific comments, properly re-arranging the introduction. Once the main scientific and technical issues will be solved, we will evaluate the opportunity of a final revision of the text by an English mother tongue expert.

REV2: 2. Geological framework - This section of the manuscript is a little bit confusing and difficult to follow. The authors jump from one topic to another in some paragraphs and is difficult to understand the geological structure of the area. I think it is necessary some organization. Begin for the big geological units, as done. Then, explain the structures, the fault systems in the area. Continue with the basins object of study. Finally talk about the seismicity in the area and the recent earthquakes and the faults that show surface rupture. In addition, I recommend the authors to be consistent with the names of the units, faults, for example, the Laga foredeep domain is referred in three or four different ways, and that is confusing.

Authors: Following the concerns of the REV2, we will reorganize this chapter, in order to introduce the concepts in a more clear and logical way, carefully checking and making consistent the names of the units.

REV2: 3. Data -The authors mention a couple of times the supporting information, but in fact the information is provided in tables and figures in the manuscript. Also, the figures in the supporting information are not correctly identified and some errors of profiles identifications are present and must be corrected.

Authors: In the supporting material we have added the high-resolution version of the figures because the pdf print considerably reduces the quality of the images, that we are conscious are important to evaluate the outcomes of our work. In addition, we have provided tables with additional information on the data, plus the conventional seismic sections in amplitude to help the readers in the comparison with the attribute images. We will fix all the issued suggested in this comment.

REV2: 4. Methods - Authors comments that they have tested several post-stack attributes, but it is not clear at all why they select ones and not others. Maybe it is not necessary to explain this? Iam not an expert in seismic attribute analysis.

Authors: Sure, we will add this explanation to the text. Meanwhile, we hope you can find an exhaustive answer also in our reply to the Jacopini's document above.

REV2: 5. Results - To me it is necessary to include in the supplementary information the profiles (original and attribute analysis) without any interpretation and each one on one page at a bigger scale.

Authors: we added the original profiles in the supplementary material, plus the high-resolution images of the interpreted attribute profiles. We'll consider the possibility to remove the interpretation from these profiles and to increase the size of the images as well.

REV2: The profiles on the manuscript show arrows pointing to specific features that attract the attention towards the author's interpretation. For example, in Fig2c the authors points with red arrows to some discontinuity (?) but at the same time the arrows mask reflectors around. I could point to similar features (orange arrow in the corresponding figure on my commented manuscript) that could point to a normal fault dipping to the W? That suggests me that the authors are just looking for structures that have been recognized at surface and not for all the other possible structures in the area/profiles. But again, without the un-interpreted profiles it is difficult to compare observations.

Authors: As remarked above, we have used this solution of the arrows to leave a certain freedom to readers in the detection of the seismic features, because we think that a standard line drawing basically tends to mask the actual seismic evidences. However, we can consider to enhance some details on the images, as a thin line drawing to highlight the main discontinuities. As discussed above, uninterpreted profiles might be added (as supplementary material) in order to allow the reader to better check of the proposed interpretation. Regarding the orange arrow proposed by the rev2, we clearly cannot exclude that there are many other similar structures. We are not only focusing on the structures visible at surface (as remarked, the Norcia antithetic fault doesn't have clear surface evidences), but simply we have specifically concentrated our study on the two basins interested by the recent seismic events (Norcia and Castelluccio di Norcia). The line CAS01 in fact, in its westernmost part, is crossing the Cascia basin, on which we did not focus our analysis. We initially thought to cut this line only showing the Eastern part (to the South of Norcia), but we finally decided to show the entire section. We'll eventually consider to cut this part during the revision if this fact confuses the readers.

REV2: I would recommend to describe each profile independently pointing to the observations done in each attribute profile and follow the same structure from one profile to the other. Begin with the seismic section and describe what you see and what is or could correspond the observed artefacts, then, the EN section with the specific observations, after, the EG section and, finally, the PR section. This makes things easy

to the reader and not necessary to jump from one profile to the other and return. I suggest to identify the different high-dipping lineaments in the figures with letters (e.g., L1, L2,: : :) and then refer to them in the text. It would be much easier for the reader to understand to which lineament the authors are referring.

Authors: Thank you for the suggestions, we'll try to do this in the revised manuscript, trying also to avoid to furnish repetitive and potentially tedious descriptions.

REV2: In profile NOR02 the relationship between horizons T, H and the west-dipping lineament interpreted as bounding the CNb is not clear. In lines 256-259 it is said that horizon H is interrupted by horizon T, which crosses all the profile from east to west and dipping to the west. Later on, in lines 275-276 it is said that a west-dipping lineament truncates and disrupts horizons (discontinuities) T and H. In general, to me is very difficult to interpret the lineaments in all the profiles (as pointed in a number of comments in the manuscript) but in that case I think that the authors are proposing different interpretations for the same observations. This needs to be clarified.

Authors: We will improve the text clarifying the relations between horizon H, the (W-dipping) low angle thrust fault T and the (W-dipping) high angle normal faults of the area, to better explain the structural complexity of this sector.

REV2: 6. Discussion and conclusions - As said in various comments I have problems to interpret the steep discontinuities on the different seismic profiles (amplitude and attributes). All the discussion is based on the authors interpretation and since I cannot interpret the same things, I cannot support it. But I am not a specialist in this type of seismic interpretations.

Authors: Following all your suggestions, as discussed above, we are confident that the proposed interpretation will result clear and possibly convincing. We will better remark in the text that vertical (and sub-vertical) features cannot be directly imaged on seismic section due to physical limits of the methodology. Such structures can be inferred considering lateral amplitude and especially phase discontinuities, which are

better highlighted by specific attributes rather than by reflection amplitude. We will improve the discussion and the images supporting our results, specifically to help the non-expert in seismic attributes to better understand the seismic features enhanced by this approach, as well as the conceptual differences and meaning.

Yours sincerely, Maurizio Ercoli and co-authors.

————————————————————

---

## Author Response (AR1)

**Marked up manuscript version**

[revised manuscript text omitted]
 using ODT software (depth conversion with VPav = 6000 m/s, vertical scale 2x).using ODT software (depth conversion with VPav = 6000 m/s, vertical scale 2x).; a) Energy+Pseudo-Relief attributes, the seismic facie in the blue box showing a strong set of deep reflectors is compared with the original seismic line in b) and Energy+Pseudo-Relief c). An analogous plot of the black box reports in figures d) and e) the original line and the combination Pseudo-Relief+Standard Amplitude.

Figure 7: Integration of surface and subsurface data (DTM by Tarquini et al., 2012); a) 3D-view (DTM by Tarquini et al., 2012) of a W-E section crossing the Norcia and Castelluccio di Norcia basins (Nb and CNb), and the mainshock locations (ISIDe working group, 2016). Surface and deep data allow to correlate the master faults and coseismic ruptures mapped at the surface. The multi-attribute display of NOR01 (b) and NOR02 (c), is obtained overlapping the reflection amplitude in transparency with the Pseudo-Relief and Energy attributes (red palette). The black boxes centred on the Norcia and Castelluccio di Norcia basins Nb and CNb have been magnified for displaying the limits of the bounding faults (black dashed lines) and the main important reflectors detected in depth. An important improvement of the subsurface images provides additional details on the seismogenic fault zones: the sketches d) and e) show an interpretation reporting the two conjugate basins, showing master faults along the borders and several ome minor synthetic and antithetic splays (see d) and e) sketches).

Figure 8: The figure proposes two alternative interpretations of the relation between the normal Vf, the deep Acquasanta thrust (T) and the Top- Basement reflector (H). Fig. 8a reports a model in which Vf merges into the deep Acquasanta thrust, suggesting a negative inversion, as a mechanism proposed by some authors (e.g. Calamita and Pizzi, 1994; Pizzi et al., 2017 Scognamiglio et al., 2018). In Fig. 8b, Vf cuts and displaces the Acquasanta thrust, following a steeper trajectory (ramp) as proposed by other authors (Lavecchia et al., 1994 and Porreca et al., 2018; 2020).

Table 1: List of some parameters extracted from SEG-Y headers and three mean frequency spectra of the three seismic lines. An approximate vertical resolution equal to 75 80 m was derived (v=6 km/s).

Fig.s1: Figure summarizing the three original seismic reflection profiles in amplitude used in this work.

Fig.s2: Figure 2 reporting the computed seismic attributes without any line drawing and labels.

Fig.s3: Figure 3 reporting the computed seismic attributes without any line drawing and labels.

Fig.s4: Figure 4 reporting the computed seismic attributes without any line drawing and labels.

Fig.s5: The image is a magnification of two portions of NOR01 and NOR02, focused on the two basins of Norcia and Castelluccio di Norcia, aiming to better display the discontinuities enhanced by the Pseudo Relief; a) PR on the Nb and interpretation of the primary (continuous lines) and secondary faults (dashed lines); b) PR on the CNb and interpretation of the primary (continuous lines) and secondary (dashed lines) faults bordering the basin.

The continuous red lines are the primary normal faults bounding Nb, whilst the dashed red segments compose a pattern of possible secondary splays within the basin.

 **Point-to-point authors response to Revision Files, by corresponding author MAURIZIO ERCOLI**

**on behalf of all co-authors.**

**Solid Earth Discussion Paper:**

**1002 **Using Seismic Attributes in seismotectonic research: an application**
**1003 **to the Norcia's Mw=6.5 earthquake (30th October 2016) in Central**
**1004 **Italy.**

Maurizio Ercoli[1;4], Emanuele Forte[2], Massimiliano Porreca[1;4], Ramon Carbonell[3], Cristina Pauselli[1;4], Giorgio Minelli[1;4],
Massimiliano R. Barchi[1;4].

---
**Colour and text code:**

**- original text (first manuscript submission)**

**-** *Rev1 and Rev2 comments: black italic*

**-** **Authors replies: blue**

---

**Manuscript Revision file – Reply to Rev1**

**REV1 General Comments:**

Ercoli et alii discuss the use of seismic attributes, applied to vintage seismic reflection data, for enhancing the structural
interpretation and faults recognition with seismotectonic purposes. They present a case study by analyzing 3 vintage lines
crossing the area interested by the 2016-2017 Central Italy seismic sequence. The study area is provided with updated
geological maps, a dense cloud of earthquake foci and some moment tensor solutions following the 2016-2017 earthquake
sequence and a dataset of earthquake-related surface ruptures, as well. This manuscript is quite well-written and the dataset
worth publication, nevertheless this work needs some major revisions, due to i) **a badly addressed paper scope**, ii) **poor**
**quality of the graphics** in their present form and iii) the somehow confusing way **the data and interpretations** are
reported.

I'm attaching an annotated version of the manuscript with many notes and suggestions; however, the major points of concern
are summarized below:

- **Data and interpretations are presented in a confusing way**. It is really difficult to follow the description of the
recognized seismic features by means of a purely qualitative pattern recognition. Graphics are not helpful in this sense and
the lack of univocal codes for e.g., faults an all the figures is making things worse. See the annotated text.

- The main point of the paper is that the use of seismic attributes can help in perform a better structural interpretation, in
particular if applied **to seismotectonic studies**. Some seismic features are here described through a **qualitative** approach and
a possible interpretation is proposed. If the main target of the work is to show the usefulness of the seismic attributes an
external dataset is needed for validation, but this is presently lacking. The use of seismic attributes allowed to identify a
**possible set of secondary structures**, near the surface, in both the Castelluccio and Norcia basins, and to propose the
presence of an antithetic fault bordering the Norcia basin to the west. Such an interpretation is not compared to detailed
geological maps (only the main structures are shown but geology is not discussed (e.g., comparing possible offset from surface geology with geophysical data). As a result, **the comparison with mapped faults** is only qualitative and quite poor. Moreover, the **seismotectonic implications of the new interpretation** is totally overlooked in the discussion and/or conclusions. In this line, I would suggest changing the title: in the present form your focusing the attention on seismotectonic research it's a really side story in the present form. A possible way to solve the lack of validation would be to make **two different interpretations**, with and without attributes, on the same dataset, basing interpretation on objective and declared principles (e.g., cutoff, peculiar seismic facies, direct fault detections, axial surfaces dying out: : : etc.) and finally compare the results with published geological maps and or sections, including the discussion on opposite interpretations in literature. - Some recent works (see a note in the text – I'm reporting here e.g., Iacopini et al. 2016 - Iacopini, D., Butler, R. W. H., Purves, S., McArdle, N., & De Freslon, N. (2016). Exploring the seismic expression of fault zones in 3D seismic volumes. Journal of Structural Geology, 89, 54-73.) proposed the use of seismic attributes for fault recognition. One of the advantages of this and other works is that you can produce a **quantitative analysis** of the wavelet, filtering out, on a statistical basis, the most probable fault plane locations. This could be helpful especially in cases where a direct detection of the seismic features is problematic. Any quantitative approach is lacking in this work: at least you should discuss the attribute range and distribution in the areas where you assume the fault should be located. I would strongly suggest **trying a quantitative approach**, at least a descriptive one. In summary, I had the impression that the aim of the work, as presently stated, is only partially achieved if an external dataset is not used for a detailed validation. Conversely, some interesting observations are arising from the Authors' interpretations: *the presence of an antithetic fault in* the Norcia basin, **the deep thick-skinned thrust** in NOR-2 section and the amount of possible distributed faults in the two basins. These points would benefit from more detailed **discussion and comparison with present proposed models in literature**. Finally, you surely have to expand the seismotectonic implications from your new interpretation. I'm sure the Authors will be able to face these criticisms and I hope that these notes will be useful to improve the present manuscript.
* * *
**Reply to general comments of REV1:**

Dear Rev1, thank you for your comments and corrections.

Following your suggestions, we have deeply revised the manuscripts, and we hope that we addressed all the main criticisms. We have also revised all the minor suggested comments, even if in most of the paragraphs have been totally rewritten in this new revised version as explained below. Regarding your main comments, we have:

i) improved the paper scope, focusing the attention on the use of the seismic attributes for a seismotectonic interpretation of the complex geological area affected by the recent seismic crisis; ii) improved the quality of the figures and graphics; iii) better distinguished the description of the data and their interpretation.

In particular,

- regarding the quality of the figures, we have improved the description of the seismic features and the graphics, that now have univocal codes (e.g. the faults line drawing and transparent polygons highlighting the interpreted fault zones) to avoid any confusion. All the main structural elements and discontinuities are now labelled and referred to the text.

- regarding the data validation, we have already remarked in the discussion phase that a validation of the data and interpretation is basically impossible in this contest: wells stratigraphy is available only in the surrounding sectors of the Apennines and not within or close to the study area. The geological complexity of this sector of the Apennines (involved at least by three tectonic phases from Jurassic to present day) does not allow to use well data, located far from the study area, to calibrate our interpretation. We have used all the geological map and stratigraphic information inferred by literature, as explained in Geological setting and Data chapters. Moreover, we have extensively used the fault patterns at surface (summarizing those main faults reported in literature) to drive the interpretation, starting from the near-surface, to link such structures to the hypocentral depth. Of course, we have then made the opposite process, drawing fault splays of fault zones where the attributes signature suggests their presence.

Using this approach, we detected the presence of antithetic fault (debated in literature) at the Norcia Basin and of a deep thrust; we also highlight the presence of some secondary faults (unmapped or not outcropping) in both the Norcia (Nb) and Castelluccio (CNb) basins, characterized by fragmented and differently oriented seismic patterns. In our opinion, the presence of fault zones makes complex and probably an excessive simplification the drawing of single fault planes, at least at the resolution provided by these data. However, we have decided to make an additional effort improving the graphics also drawing, as suggested, some possible faults alignments in a new figure to better explain the interpretation process and criteria used. Where the high-dipping discontinuity (mainly in phase and/or amplitude) were separating different reflection patterns and truncating reflectors, we have added a primary fault (continuous red lines and polygons). When similar but smaller discontinuities between reflectors were particularly evident, parallel or antithetic to the principal faults, we have added a fault splay/secondary fault. A more quantitative approach as well as an estimation of the offset based on such data is difficult to achieve, therefore we have rewritten as suggested the description on the attribute performance in the areas where we think the faults are located.

Regarding the discussion part of the paper, we have improved the seismotectonic implications with respect to the models debated in the literature. In particular, we have defined the main potential seismogenic faults at depths (e.g. Norcia and Vettore faults) and discussed the relationships of active normal faults and inherited structures highlighted by attributes analysis. In this latter case, we have proposed two different interpretations of the cross-cutting relationships between the seismogenic Vettore fault and a deep thrust (see last part of the chapter 6), as suggested by the Reviewer. We have also added a new figure (Fig. 8) to describe and compare these two models.

Taking into account all these improved arguments on the seismotectonic features of the area, we have finally decided to keep the same title, focused on the seismotectonic implications of the seismic active area of the Apennines.

**Manuscript Revision file – Reply to Rev1 supplement:**

Lines 19-20:

*REV1: Rather than this quite general sentence, insert one sentence summarizing the methods of analysis here adopted.*

Authors: We have added short info on the attributes used, then we move forward the sentence, to reinforce the outcomes about the detection of faults currently debated in literature (e.g. the Norcia w-dipping antithetic fault).

Line 27: Introduction

Line 27: *shorten up the introduction avoiding repetitions and trying to better focus on the topic of the manuscript.*

Authors: we have rewritten and shorten the introduction chapter, trying to improve the text and better focusing the main topics, as requested.

Lines 29-31: "Clearly, this is not an easy task: it is in fact generally complex to fill the gap between the exposed geology including the active "geological faults" mapped by the geologists and the seismic features describing the geometry"

*REV1: you made a big jump in the logic here. You are already focussing on seismic reflection data while there is a bunch of other techniques. you described some approaches later in the text but you should move that part here, I suppose.*

Authors: we have corrected and rewritten this sentence, introducing first the other geophysical techniques.

Lines 38-39: "This fact generates uncertainties that may amplify the scientific debate and the number of models introduced by the geoscientists. Therefore, this process requires the use of appropriate geophysical data, aimed at recovering information on the deep geological architecture and, in particular, on the geometry of active faults."

*Rev1: This statement is arguable: the aim should not be to obtain a consensus on interpretations but to provide as many constraints to interpretations as possible.*

Authors: we agree with this comment. We have rewritten this sentence focusing the attention on the use of the seismic attributes to improve the subsurface geological interpretation and to achieve additional information from the 2D data. The final aim is to obtain constraints on the geological structures responsible for the seismicity of the area, and in particular to define geological/structural setting at depth (e.g. depth of the basement and its involvement) and to trace of potentially seismogenic faults.

Lines 57-58: "To improve the data quality and increase the accuracy of the interpretation, three main strategies can be usually considered: 1) collection of new reflection seismic data with modern technologies, optimizing feasibility studies on the base of available vintage datasets;"

*Rev1: this is partly already stated at lines 45-46.*

Authors: we have deeply reorganized and rewritten the text, removing possible repetitions.

Lines 63-65: "Some limitations characterize the first two approaches: the first is particularly demanding in terms of costs and logistic, and not practicable in zones where the use of dynamite or arrays of vibroseis trucks is forbidden or limited (e.g. National Parks or urban areas)"

*Rev1: also this is already introduced at lines 45-46. try to sum up the three parts.*

Authors: we have modified this part to avoid repetitions, as requested.

Lines 92-98: "After the last 2016-2017 seismic sequence, Porreca et al. (2018) have provided a new regional geological model based on the interpretation of vintage 2D seismic lines. In such a study, the authors remark important differences in the seismic data quality across the region. In fact, the eastern area that shows higher overall data quality, is located at the footwall of the Mount Sibillini thrust (MSt) and, includes (consists of) flyschoid units of the Laga foredeep Domain. It is noteworthy that the Mw 6.5 epicentral zone, is located on the MSt hanging-wall (Lavecchia, 1985). This is characterized by prevalent carbonate sequence and, its crossed by seismic sections with lower S/N ratio, that hampered the subsurface interpretation."

*Rev1: move this part from the introduction to the geologic framework*

Authors: we have moved this part to the geology chapter. This latter has been extensively re-organized as suggested also by Rev2.

Lines 100-101: "The main goal of this study is to obtain as much information as possible on the geological structures responsible for the seismicity."

*Rev1: try to rephrase. the aim is not clear. could you better explain what characteristics of the seismogenic source are you going to better define thanks to your analysis?*

Authors: we have rephrased the sentence improving the main aims of the study. See the response above.

Lines 103-104: "The current manuscript is an example of how can seismic attribute analysis contribute to seismotectonic research as an innovative approach."

*Rev1: this should be rephrased. limiting the impact of this work to a simple case study is not promising and adequate to this journal. The importance of this work could be by far better underlined if you clearly state from the very beginning the different interpretations postulated on the Central Italy seismogenic structures and your contribution on this open debate. The introduction should be mostly rewritten in this sense: at the moment there is a general overview on attribute analysis and you end up by proposing a case study.*

Authors: we totally agree with this comment as it was the aim of this work. We aim to present not only a case history, but we want suggest this approach as a valuable solution for seismotectonic studies around the world. Thus, we have improved and rewritten the introduction, trying to better explain the contribution of this study to seismotectonic interpretation of the area. We refer to Porreca et al. (2018) in the geology chapter for the different interpretations postulated about the Central Italy seismogenic structures.

Lines 108-109: " "

Authors: we have entirely rewritten the chapter. This sentence also has been modified, just to remark the importance of the 2016-2017 sequence.

Lines 113-114: "… belt, including the Umbria-Marche thrust and fold belt domain and Laga Formation."

*Rev1: add a REF here and introduce to international readers a brief sentence summing up the meaning of Umbria Marche and Laga Fm. significance.*

Authors: we have modified the sentence adding some references.

Line 120: "…faults since the Late Pliocene"

*Rev1: add a ref here*

Authors: done.

Line 122: "sequence"

*Rev1: you were referring to Laga Fm. above. be consistent.*

Authors: done. We now refer to Laga sequence.

Line 124: "velocity (Vav = 4000 m/s)"

*Rev1: you were referring to Laga Fm. above. be consistent.*

Authors: we have rewritten the text and fixed these issues.

Lines 142-150: "…Norcia (Nb) and Castelluccio di Norcia basins (CNb) (Fig. 1). Nb and CNb are..."

*Rev1: are all these acronimous really necessary? cue them when possible. e.g., Nb and CNb can be probably deleted.*

Authors: we agree that in this section there are many acronyms, but we have decided to maintain in particular Nb and CNb, also following a Rev2 comment. They help to shorten the document and are useful to refer them to the figures.

Lines 178-180: "…OpendTect (OdT) software… QGis software… from maps and Ithaca database"

*Rev1: add the project URL, which maps? add the REF and project URL*

Authors: we have added the URL and removed "maps" (already listed in the next raw) rewriting the sentence.

Lines 192-193: "(Barnes 1996; Taner et al., 1979; Barnes, 1999; Chen and Sidney, 1997; Taner, 2001;

Chopra and Marfurt, 2007; Chopra and Marfurt, 2008; Forte et al., 2016)"

*Rev1: there is some other and more recent literature to be cited... e.g., Iacopini and Butler, 2011;*

*Iacopini et al., 2012; McArdle et al., 2014; Botteret al., 2014; Hale, 2013 for a review; Marfurt and*

*Alves, 2015*

Authors: we have added the recent literature, as requested.

Line 200: "Energy" (E):

*Rev1: it would be better to provide a generalized formula, at least for this attribute.*

Authors: We added a reference in the text, referring to a specific paper of our co-author Emanuele

Forte, in which all the mathematical formulation is already provided within an exhaustive appendix.

Line 206: "…useful to emphasize the most reflective zones…"

*Rev1: provide a reference to the software used for attribute calculations.*

Authors: reference are added in the text.

**5. Results**

Line 226:

*Rev1: the reporting of the results in quite confused. There is eccessive use of acronymous, text jumps*

*from continuously from one sector to another making the reading very frustrating. More importantly,*

*the text does not highlight the advantages and limitations of each technique. You should provide a first*

*interpretation of faults, based on geological data and amplitude sections, and then provide a refined*

*interpretation using seismic attributes. This approach would stress the real advantages of using seismic*

*attributes.*

Authors: We agree that there are some acronyms, but we have maintained most of them because the text would be even worse by repeating the long names of basins and faults (also following the advice of

Rev2 to continue using Nb and CNb once defined). Then, we have reorganized the chapter improving the text and adding boxes/labels for interpretation of the amplitude section, poorly informative regarding the faults.

Line 238: seismic profile, and in addition it is partially interfering with suspicious processing artefacts (highlighted with yellow dots, labelled as "A", slightly undulated in Fig. 3a whilst horizontal in Fig. 2a ca. at 1 s)

*Rev1: discuss this artefact. where is it coming from?*

Authors: the legacy seismic lines have been provided already processed by ENI, so we suspect this is the result of a windowed filter to remove horizontal noise or multiples. We have added this consideration within the text and we have marked these artefacts in the figures.

Line 243: "…by the EG and PR attributes (Figs. 3c and 3d) …"

*Rev1: data description is quite confusing: try to label each feature with letters on the seismic lines*

*instead and refer to those codes.*

Authors: the acronyms for the main faults are provided on the top of the PR attribute, whilst letters are provided for the low angle features (H and T) (and blue/black boxes). We avoided to add extra letters and labels for the secondary faults to within the text and figures that are already dense of items.

Line 264-266: In fact, a main high-angle E-dipping discontinuity (red arrows) delimits the NOR02

western sector (ca. 1 km of distance along the line at surface); another steep W-dipping alignment (red arrows) clearly cuts and slightly disrupt the set of reflectors below the Nb (0-2.5 s, ca. 4-5 km).

*Rev1: there is a plenty of red arrows in Fig. 4 c and d. It is really hard to follow such a description.*

*Maybe provide a letter for each element whose you are referring to in the text.*

Authors: as remarked in the comment above, we avoided to add more letters, the arrows indicate the main areas in which the discontinuities are visible. We have added transparent red polygons the help the readers to focus on the main discontinuity areas. We have also improved the quality and brightness of all the figures, and added an extra figure (s5) with two zooms on two areas to better show the alignments and better clarify the interpretative strategy and criteria.

Line 268: fragmented reflectors pattern in the middle portion.

*Rev1: there is no line drawing of these secondary elements in Fig. 4 c and d. Instead, in the Norcia*

*basin (kms 0 to 5) some gently W-dipping reflectors can be traced, probably indicating backtilting to*

*the west of this crustal sector. If this is true, the backtilting could possibly indicate that the main fault is*

*the E-dipping one (see also Fig. 7). could you discuss this observation or discard this hypothesis?*

Authors:   In our opinion the W-dipping reflectors (we agree that these are the most evident features in the seismic profile) derive from the SW-dipping tectonic units, so they are mostly related to compressional tectonics. But in particular, in this sentence we wanted to highlight the fragmentation of these reflectors created by a dense pattern of subvertical discontinuities suggesting the presence of a fault zone (shown by a peculiar signature of faulting on these seismic lines). Instead of a single fault lineament we prefer the concepts of fault zone made by many steep secondary discontinuities and fragmented fabric concentrated in a narrow area. Following this consideration, we used first only some aligned red arrows to drive the readers' attention on the main discontinuity zones. Then have introduced also an additional figure (s5) as requested, with two magnifications on representative areas illustrating a simple interpretation of the most visible faults.

Line 276: "seems"

*Rev1: try to avoind the term "seem" and similar. It gives the feddback that your new imaging is not*

*reducing the uncertainties. Moreover, in the data and results section, only objective information should*

*be given.*

Authors: ok, removed in the entire part.

Line 282: …combined plot of the PR attribute…

*Rev1: "the multi-attribute rendering method should be introduced in "Methods"."*

Authors: "multi-attribute display" was already in "Methods", but we have changed it now with

"rendering" as requested and added a specific reference.

Line 282: "("similarity" palette) with superimposed the EG attribute ("energy" palette)"

*Rev1: this is not clear, what do you mean with superimposed? a transparency? or a multi-band false*

*color rendering? the first I suppose.*

Authors: transparency, corrected.

Line 283: "(depth conversion with $V_{Pav}$ = 6000 m/s, vertical scale 2x)."

*Rev1: sorry but I'm missing this point... could you be clearer?*

Authors: deleted, it was a mistake.

Line 285: "The blue box of Fig.5a is reported in Fig. 5b and 5c by…"

*Rev1: this should go in the figure caption.*

Authors: text has been changed according to this.

Lines 294-296: "Such results therefore ensure an easier and more accurate interpretation of the subsurface geological structures; those are connected with the surface geology and related to the hypocentre location of the main seismic events, that will be discussed more in detail within the following chapter.

*Rev1: (divided in some points)*

*1) no interpretations are given for these figures. In order to demonstrate the supposed enhancement you*

*should provide a line drawing with horizons, cutoffs etc. on each rendering, demonstrating and*

*discussing which seismic features are better imaged through each rendering.*

*2) how can you assess that you are correctly interpreting the signal?*

*3) Is there any geologic evidence such as the 2016 ground breaks?*

*4) can you compare your sections with detailed fault strand traces after recently published geologic*

*maps? e.g., Pierantoni et al.?*

Authors:

1) Instead of using a standard line drawing we have used boxes, dashed lines and arrows to leave the sections cleaner for readers to see the improvements. But we have also added the figures 2s,3s,4s displaying the attributes without any interpretation labels as requested by the second reviewer, and also adding a new figure (5s) displaying the interpretation of two representative basins. In addition, our final interpretation has been summarized in figure 7, in a discussion considering all the other data available for the area including outcropping geological units (carbonate substrate vs. quaternary basins), the main faults and the surface ruptures (point 3). Finally, we have improved the figures drawing some boxes, lines, labels etc ...  enhancing the features displayed by the attribute analysis.

2) We remarked that the only constraints available are at the surface (geology and traced faults), so our seismic interpretation is clearly based on our experience and knowledge of the Central Apennines, from the geologic and geophysical point of views. On the other hands, the geophysical features are interpreted using common and well-known principles available in literature, particularly regarding the signature of faulting. However, for the interpretation of the deepest (less-constrained) part of the seismic images, we have produced a new figure (Fig. 8) reporting two different interpretations as suggested by the Reviewer.

3) Surface evidences can be observed in the field and there is a wide literature cited in this work, like co-seismic ruptures (e.g. Civico et al. 2018, Villani et al., 2018a, Brozzetti e al., 2019 and many others). Not only geomorphological and geological evidences, but also paleoseismological data (citations in the text). Surface ruptures have been observed in the Central Apennines area, also in the past, only after earthquakes of Mw > 6.

4) this is what we aimed to do in this work, but probably unclear in the first manuscript version. However, in this revision we have better separated in the text and figures the surface data (including known faults and surface ruptures, detailed in Fig.1) by our fault interpretation.

We basically have started our seismic interpretation using the surface data, therefore "driving" our workflow using the location of the known faults and ruptures at surface. Secondly, by considering "peculiar signature of faulting" obtained by attributes computation, we interpreted other buried faults, fault zones or secondary splays. The best example, among our results, is the detection of a primary fault still debated in literature due to scarce surface evidences: it is the Norcia antithetic fault, that in our opinion is "seismically" very clear in our attribute sections.

Lines 310-312: "The deep, high-amplitude reflector (H, blue arrows and dashed line) highlighted to the West of Nb in NOR01 (at 2.5 s, in Figs. 2d and 7d and in Figs. 3d of CAS01), presents an attribute signature similar to the one deeper visible in NOR02 beneath CNb (3.2 s, in Figs. 4b and 7e)."

*Rev1: this is a repetition..."*

Authors: the aim of this sentence was to correlate and group the observation done for the H reflection visible in NOR01(and CAS01) with NOR02, that was not the objective of the previous chapter. However, we have rewritten the text, particularly focusing on the interpretation aspect.

Lines 313-315: "This set of reflectors are interpreted as a high acoustic impedance contrast, possibly related to an important velocity inversion occurring between the Triassic Evaporites (anhydrites and dolostones, $Vp \approx 6$ km/s, e.g. Trippetta et al., 2010) and the underlying acoustic Basement (metasedimentary rocks, $Vp \approx 5$ km/s, sensu Bally et al., 1986)."

*Rev1: this interpretation implies that the Sibillini thrust is thick-skinned. this is an important consequence of your interpretation. try to stress it in the discussion.*

Authors: We have discussed to possible scenario for the interpretation of the deeper discontinuities and reflectors. In both cases we are not able to resolve the duality between thick- and thin-skinned tectonics. We have described this in the Discussion chapter.

Line 323: "(Figs. 2d and 7d)"

*Rev1: why are you not using the codes in Figs? this paragraph is really confusing. try to rewrite it with the help of univocal codes for surface geology and seismic sections...*

Authors: we have rewritten the paragraph, using the codes introduced for the surface geology/faults and seismic sections.

Line 351: "Those"

*Rev1: ???*

Authors: corrected

FIGURES

REV1

Figure 1:

*Rev1: Provide the codes for the faults reported in sections. Are these all the potentially active faults reported in geological maps or a selection of?*

Authors: We have added the codes for the main faults bounding the basins. We provide, after a comprehensive literature review, a summary of all the main faults and secondary splays mapped on the area.

Figure 2:

*Rev1: 2C -> these features in red are not well detectable. maybe you should use a more quantitative approach to characterize them. e.g., semblance coherence or other quantitative measures of attribute similarity...*

*blue on green is not a good choice for the readability indicate H also here.*

Authors: due to the nature of the data, we have declared that this study has a qualitative approach, being such results the best we are able to provide. The tests performed with other attributes like the similarity didn't perform well (see our reply the reply to Prof. Iacopini during the discussion).

However, the Norcia antithetic fault looks clearer in comparison to the Norcia fault. The position of Nf is constrained by surface outcrops, but also looking at all the three attributes (particularly the PR, better showing the changes in the reflection patterns) it is plausible in this position and with this geometry, and suggesting a deformation spread in a narrow fault zone.

We have updated the figure as requested, modifying the arrows for better visibility.

Figure 3:

*Rev1: 3A -> CHANGE THE COLORS IN ORDER TO INCREASE CONTRAST. provide a colorscale for the use palette: what is the range of values of each attribute?*

Authors: We have increased the images contrast and added the colour bars as requested.

Figures 5-6:

*Rev1: in the main section report the letters for the insets... the fault from surface geology, in red, and their codes are not readable....*

Authors: we have used a colour code (blue and black) thicker on the boxes. We have improved the fault labels at surface and the overall quality of the attribute images.

Figure 7

*Rev1: these beachballs have not been projected onto the 3D perspective. it could be misleading... you can simply report them in sections as done for the Mw 6.5 event.*

Authors: apart the Mw 5.3 event very close to the line, the other events a too far for a reliable re-projection on the sections. So, we left only the beachball of the mainshock (rotated considering the perspective).

Line 693: "EN+PR"

*Rev1: expand the codes in the caption....*

Authors: fixed

Final comment:

We have produced a new figure in the main text (Fig.8) proposing two possible interpretations of the cross-cutting relations between deep reflectors, normal faults and thrust. We have also improved and added new figures in the Supplementary as requested during the first revision.

**Manuscript Revision file – Reply to Rev2**

**REV2 General Comments:**

The manuscript "Using Seismic Attributes in seismotectonic research: an application to the Norcia's Mw=6.5 earthquake (30th October 2016) in Central Italy" by Maurizio Ercoli et al. submitted to Solid Earth proposes the use of seismic attribute analysis approach on three vintage reflection seismic profiles acquires across the Norcia and Castelluccio di Norcia basins to determine the extension and geometry of the geological structures. This region was the epicentral area of the 2016-2017 seismic crisis in central Italy. This manuscript could be of interest to geologists and geophysicists working in active tectonics and using reflection seismic data. However, in my opinion, it needs still some work in the structure of the writing and, most important, more work in the interpretation of the data or, at least, it needs to show more clearly all the interpretations the authors are doing. I am not an expert in the analysis of this type of data (onshore seismic data across rocky regions) but I have many difficulties to identify the same structures the authors are interpreting. At the end, I have had the impression that the authors have extended the surface map structures in depth following some possible alignments. My question is, would have they interpreted the same structures without the surface information? To me, there is a **high uncertainty in the interpretation of the alignments** in the seismic profiles that, then, I have problems to believe the final structural model proposed in the manuscript. Following there are some general comments on the different sections. I also provide a commented manuscript that hope will help to improve the quality of the manuscript and the presented results. Despite my criticism, to be intended solely as constructive, I warmly encourage the **authors to make any effort for the publication of this manuscript**, because of the relevance of the proposed approach and objectives.

1. **Introduction** I think that in general the introduction needs to be restructured to emphasize the main aspects of what authors wants to expose. It is a very confusing introduction. I am not a native English speaker and I have found some errors, so I think that a native English speaker should review the final version of the manuscript. Some specific comments: Paragraph from lines 69 to 104 is a long paragraph that jumps from one idea to another and then back on. It is confusing and needs to be rewritten. Why mention 2D data vs 3D data various times? Just need to stress the differences and then stress the information and advantages of using 2D dataset, mainly which it is available and ready to work on. In addition, sentences like the one in lines 82-84 are out of sense in that paragraph. The stated between lines 85 and 98 is confusing. This may be rewritten, but also, I think that it makes no sense to explain all this in the introduction.

2. **Geological framework** This section of the manuscript is a little bit confusing and difficult to follow. The authors jump from one topic to another in some paragraphs and is difficult to understand the geological structure of the area. I think it is necessary some organization. Begin for the big geological units, as done. Then, explain the structures, the fault systems in the area. Continue with the basins object of study. Finally talk about the seismicity in the area and the recent earthquakes and the faults that show surface rupture. In addition, I recommend the authors to be consistent with the names of the units, faults, for example, the Laga foredeep domain is referred in three or four different ways, and that is confusing.

3. Data The authors mention a couple of times **the supporting information**, but in fact the information is provided in tables and figures in the manuscript. Also, the figures in the supporting information are not correctly identified and some errors of profiles identifications are present and must be corrected.

4. Methods Authors comments that they have tested **several post-stack attributes**, but it is not clear at all why they select ones and not others. Maybe it is not necessary to explain this? I am not an expert in seismic attribute analysis.

5. Results To me it is necessary to **include in the supplementary information the profiles** (original and attribute analysis) without any interpretation and each one on one page at a bigger scale. The profiles on the manuscript show arrows pointing to specific features that attract the attention towards the author's interpretation. For example, in Fig2c the authors points with red arrows to some discontinuity (?) but at the same time the arrows mask reflectors around. I could point to similar features (orange arrow in the corresponding figure on my commented manuscript) that could point to a normal fault dipping to the W? That suggests me that the authors are just looking for structures that have been recognized at surface and not for all the other possible structures in the area/profiles. But again, without the un-interpreted profiles it is difficult to compare observations. I would recommend to describe each profile independently pointing to the observations done in each attribute profile and follow the same structure from one profile to the other. Begin with the seismic section and describe what you see and what is or could correspond the observed artefacts, then, the EN section with the specific observations, after, the EG section and, finally, the PR section. This makes things easy to the reader and not necessary to jump from one profile to the other and return. I suggest to identify the different high-dipping lineaments in the figures with letters (e.g., L1, L2) and then refer to them in the text. It would be much easier for the reader to understand to which lineament the authors are referring. In profile NOR02 the relationship between horizons T, H and the west-dipping lineament interpreted as bounding the CNb is not clear. In lines 256-259 it is said that horizon H is interrupted by horizon T, which crosses all the profile from east to west and dipping to the west. Later on, in lines 275-276 it is said that a west-dipping lineament truncates and disrupts horizons (discontinuities) T and H. In general, to me is very difficult to interpret the lineaments in all the profiles (as pointed in a number of comments in the manuscript) but in that case I think that the authors are proposing different interpretations for the same observations. This needs to be clarified.

6. **Discussion and conclusions as said in various comments I have problems to interpret the steep discontinuities** on the different seismic profiles (amplitude and attributes). All the discussion is based on the authors interpretation and since I cannot interpret the same things, I cannot support it. But I am not a specialist in this type of seismic interpretations.
* * *
**Manuscript Revision file – Reply to Rev2 supplement:**
Dear REV2, thank you for all your detailed comments. In this new revised manuscript, we have improved, as suggested, the data interpretation in the text and the figures to show more clearly all the interpretations that we propose. We agree that an attribute analysis done for seismotectonics is a new and complex approach for non-experts, particularly on onshore vintage data like the ones reported in this work. But we aim to give some slight improvements (possibly not fantastic like in offshore 3D seismic volumes) supporting the data interpretation. We aim to suggest to scientists working on such topic a new approach able to achieve better constraints on seismic areas characterized by scarcity of deep data. To do this, we have declared at the beginning of the manuscript that our strategy is based on the extension of the surface map structures in depth by following some possible seismic alignments, as the geologic data at surface are the only constraints available (absence of deep wells stratigraphy).

Regarding the main points:

1) The introduction has been completely rewritten following all the suggestions and the correction of both reviewers. In particular, we have shortened it and better focused the aims of this work as explained above (please see also responses to Rev1).

2) The geological framework has been totally reorganized and rewritten in a more logic way, using the scheme proposed by Rev2.

3) The supporting material contained the raw seismic lines plus the high resolution (pdf) images of the attributes, effectively with some possible mistakes in the filenames. However, we have entirely reorganized the material. Now we have added 5 figures to the Supplementary material: fig.1 summarizes the original lines, the figs. 2s, 3s, 4s reports the attributes without labels as requested for better comparison, fig.5 is finally another figure regarding the details of the PR attributes and their interpretation, related to the two tectonic-controlled Quaternary basins.

4) We have improved this paragraph briefly describing the workflow done to select the attributes. Further details have been provided during the discussion phase in the reply to Iacopini, but later we have inserted in the manuscript only a summary. This is to avoid an excessive technical description which in our opinion would have distracted the reader from the main theme of the work.

5-6) We have included in the supplementary information the original amplitude profiles as well as the attribute analysis without interpretations (point 3). We have also remarked in the text that we looked for structures that have been recognized at surface. We started our interpretation using this constraint at surface, but then we extended the interpretation to the geophysical signature of faulting also belonging to possible structures not outcropping in the area/profiles (mainly the two basins of Norcia and Castelluccio di Norcia). We have rewritten the text following the Rev2 advice, even without grouping similar observations to avoid boring repetitions. We have better labelled at least the main structures (aNf, aVf, Nf, Vf), even if without labelling each secondary splay to avoid an excessive use of the acronyms/labels in the text (note by Rev1).

**Manuscript Revision file – Reply to Rev2 supplement:**
Lines 16: …recently…

*Rev2: Recently is an ambiguous term. Instead, you could include the time range of the seismic sequence.*

Authors: we agree with this comment, we have added the time range 2016-2017, as requested.

Lines 18: … …

Authors: we decided to maintain this sentence but adding "at the regional scale" because such data are the only available, so we'd like to remark their importance.

Lines 34: … impressive topographic changes…

*Rev2: Consider to delete.*

Authors: removed and changed with "important"

Lines 36: … While many studies on the surface geology are generally performed, especially after important events …

*Rev2: I do not agree with this. There have been studies of active faults around the world before the occurrence of a large earthquake, not just after. In fact, I would say that is on the contrary, a lot of faults have been studied that do not have produced an earthquake nor in recent or historical times.*

Authors: we have entirely rewritten the introduction. See main comments.

Lines 38-40: … This fact generates uncertainties that may amplify the scientific debate and the number of models introduced by the geoscientists. Therefore, this process requires the use of appropriate geophysical data, aimed at recovering information on the deep geological architecture and, in particular, on the geometry of active faults.

*Rev2: I have understand what authors want to express with this sentence after read it few times. Recommend to rewrite. Which process? Obtaining? Adquireing?*

Authors: we have entirely rewritten the introduction.

Lines 42-49: … Different geophysical methods (e.g. Gravimetry, Magnetics, Electric and Magnetotellurics, Ground Penetrating Radar) may contribute to define the stratigraphy and structural setting of the upper crust at different scales. But the seismic reflection is largely the most powerful tool producing high-resolution images fundamental to trace the actual geometry of active faults at surface (usually mapped and reconstructed in geological cross-sections), from the near surface down to hypocentral depths. However, the ex-novo acquisition of onshore deep reflection data, possibly 3D, is often hampered by environmental problems, complex logistics, and high costs. These issues seriously limit the possible, widespread use of this technique for scientific research. Significant exceptions are research projects for deep crustal investigations like BIRPS (Brewer et al., 1983), CoCORP (Cook et al., 1979), ECORS (Roure et al., 1989) and CROP (Barchi et al., 1998; Finetti et al., 2001), IBERSEIS (Simancas et al., 2003).

*Rev2 (grouped questions): is the method that provides...? Confusing, rewrite. Is this necessary?*

*Nowadays seismic acquisition is extensively used, although I agree that it is being more difficult to*

*acquire deep seismic data, but it is still possible. Is that necessary? I know that some research groups in*

*France and Spain have acquired deep seismics (reflection and refractions) in the Mediterranean in the*

*last decade, so in more recent times that all these other datasets.*

Authors: the entire paragraph was rewritten and recent references updated as requested.

Lines 50-51: …Such limitations can be partially overcome by considering old profiles (legacy data)

acquired by the exploration industry. When collected in seismically active regions, such data may be used to connect the active faults mapped at the surface…

*Rev2: I am not in agreement with this statment, I think that even a little more difficult it is not*

*impossible to acquire new seismic data. I think that the use of legacy data could be a nice source of*

*data in places that new data is difficult to acquire due to lack of funding or that could provide new*

*information to improve the geological models. I think that you try to justify the use of legacy data*

*pointing to limitations instead of pointing to advantages, as would be the already availability of these*

*data. I would consider to rewrite this part.*

Authors: We actually agree that it is not impossible, we have just remarked that currently it is not common to see research projects including acquisition of regional seismic reflection data for seismotectonic purposes. More common is the acquisition of high-resolution seismic at the scale of single basins. We appreciate the advice regarding a justification for using legacy data considering the advantages instead only the limitations. So, we have rewritten the introduction following this indication as requested.

Line 51: …such data may be used to connect the active faults…

*Rev2: to improve geological models... Usually researchers working in seismotectonics has tried to do*

*that link between surface geology and earthquakes proposing different fault models, isn't it? The Italian*

*active faults database localize active faults provide fault dip, seismogenic depth, so it defines a fault*

*model for each source. Your data may improve the determination of the fault geometry and other*

*characteristics.*

Authors: We totally agree with this comment. We have specified that the results of this approach can be useful for constraining the subsurface geological setting and to provide new data on active tectonic structures. We have also cited the DISS database (Basili et al., 2008) as an example of database of active faults in Italy.

Line 57: … three main strategies can be usually considered…

*Rev2: Where? In seismic processing?*

Authors:  we have rephrased the sentence: "In order to improve the data quality and increase the accuracy of the interpretation, two main strategies, ordinarily used by the O&G industry, can be applied on legacy data: 1) reprocessing from raw data using modern powerful capabilities, processing strategies and developments of newly performing algorithms and software; 2) use post-stack analysis techniques such as seismic attributes."

Lines 66-67: … The second requires broad projects encompassing specialized teams, high-computation power and generally long processing times, the latter is dependent on the quality of the raw data. The third strategy, in the case of the attribute analysis exploits a well-known and mature technique…

*Rev2 (grouped questions): That is not true. I agree that it is a time consuming task and maybe you may need a dedicated workstation, but reprocessing seismic data does not requires a broad project and large teams.*

*I do not understand this sentence, at the beginning I thought you were describing the third type of strategy, just after I have seen I was wrong. Rewrite*

Authors: we have rewritten and simplified the entire paragraph, following the advices of both reviewers. Regarding the costs, time, and team availability, the problem is wide and complex to be fully described here. However, with "modern processing techniques" we meant specific type of workflows e.g. including Pre-Stack Depth Migration (PSDM), that may require high computational power, long time and teamwork if performed on densely sampled 2D lines and/or 3D data in a short time period. Currently, only the oil companies or their contractors have such possibilities, whilst clearly, it's less easy, even if not impossible, in academic environments. Of course, we agree that more conventional workflows, depending on the survey goals, can be accomplished with more limited efforts.

Line 72: … seismic volumes produced spectacular results…

*Rev2: This is ambiguous. Could you describe very briefly these results or give a couple of examples? For example: "...volumes allow identifiyng ancient river channel and ..." in agreement with your citations.*

Authors: thank you for the suggestion. We have integrated the text as requested.

Line 77: … in complex geological areas …

*Rev2: Just in complex geological areas? Conisder to delete.*

Authors: We agree with your comment. We have modified the sentence as: "... the attribute analysis is probably the easiest, cheapest and fastest to qualitatively emphasize the geophysical features and data properties of reflection seismic data sets, producing benefits particularly in complex geological areas."

Line 79: … may not bring so impressive improvements …

*Rev2: may not provide the same quality of information than on 3D*

Authors: corrected

Lines 79-81: … However, the main point is that inland, most of the sedimentary basins have actually been sampled by 2D grids of seismic profiles, or at least they have been probed by a few sparse 2D seismic lines.

*Rev2: Maybe that is your case, but it could be not the same thing in other areas. I would rewrite this sentence pointing that you use this data because it is the available data.*

Authors: We rephrase as follow: "However, the main point is that in the past, it was common to sample study areas inland by 2D grids of seismic profiles, being the full 3D seismic surveys rare"

Lines 82-84: … Whilst in the hydrocarbon industry this process is useful even if mainly driven by a constant necessity to reduce the costs (Ha et al., 2019), in seismotectonic researches it is affected by even worse limitations previously aforementioned …

*Rev2: Consider to delete.*

Authors: we have cancelled this sentence as requested

Line 87: … Based on such considerations,…

*Rev2: Which ones?*

Authors: Deleted

Line 90: … proposed new approach …

*Rev2: Which new approach?*

Authors: we have rewritten the text explaining better which approach we propose.

Line 96: … Mount Sibillini thrust (MSt) …

*Rev2: Indicate it in figure 1*

Authors: MSt added in Fig.1

Lines 103-104: … The current manuscript is an example of how can seismic attribute analysis contribute to seismotectonic research as an innovative approach

*Rev2: This is a conclusion.*

Authors: we have rewritten and integrated the sentence following the comments of both reviewers.

Line 108: … L'Aquila and Colfiorito, …

*Rev2: Indicate the years of the events*

Authors: years added in the text.

Line 109: … 97'000 events …

*Rev2: ?*

Authors: it was the total number of earthquakes recorded in two years. However, we have rewritten the sentence following the comments of both reviewers.

Lines 111-112: … generating impressive co-seismic ruptures (Civico et al., 2018; Brozzetti et al.,

2019)...

*Rev2: Necessary? Where? Along the Mt Vettore fault? Also point to Fig1*

Authors: corrected.

Lines 113-128: The study area is located in the easternmost part of the Northern Apennines fold and thrust belt, including the Umbria-Marche thrust and fold belt domain and Laga Formation. This is a geologically complex region, where in the past the analysis of 2D seismic profiles have produced contrasting interpretation of the upper crust structural setting, e.g. thin vs. thick skinned tectonics, fault reactivation/inversion, basement depth (Bally et al., 1986; Barchi, 1991; Barchi et al., 2001; Bigi et al.,

2011; Calamita et al., 2012; Porreca et al., 2018). The Umbria-Marche fold and thrust belt was formed during the Miocene compressive phase, and overthrusts the Laga foredeep sequence, through arcshaped major thrusts, namely the M. Sibillini thrust (MSt, Koopman, 1983; Lavecchia, 1985), with eastward convexity. The compressional structures were later disrupted by the extensional faults since the Late Pliocene. The Umbria-Marche domain involves the rocks of the sedimentary cover, represented by three main units:

1) on top, the Laga sequence consisting of siliciclastic turbidites belonging to the Laga foredeep and foreland Formation (Milli et al., 2007; Bigi et al., 2011); it is made by alternating layers of sandstones, marls and evaporites (Late Messinian – Lower Pliocene, up to 3000 m thick, average seismic velocity (vav) = 4000 m/s), mainly outcropping in the eastern sector of the study area (i.e. at the footwall of the MSt).

2) in the middle, carbonate formations (Jurassic-Oligocene, about 2000 m thick, vav= 5800 m/s) formed by pelagic limestones (Mirabella et al., 2008) with subordinated marly levels overlying an early Jurassic carbonate platform (Calcare Massiccio Fm.)

*Rev2 (grouped questions): Identify in Fig.1 "Umbria-Marche thrust and fold belt domain". You identify it as Laga foredeep domain in Fig1, later as Laga foredeep sequence and here as Laga Formation. Be consistent and use the same terminology along the manuscript and figures.*

Lines 131-132: representing the main ad deeper detachment of the region.

Line 133: An underlying basement of variable lithology (Vav = 5100 m/s)

Line 135: aforementioned units by the aforementioned important regional decollement.

Lines 136: … complex …

*Rev2: represents or is where the detachments are localized?*

*Rev2: Rewrite*

*Rev2: Repetitive and ambiguous. Rewrite.*

*Rev2: I wouldn't say complex, is just a quite simple thrust and fold system, isn't it?*

Authors: all these corrections have been considered and this paragraph has been totally rewritten.

Line 137: … produced NNW-SSE striking WSW-dipping normal faults …

*Rev2: All the faults are dipping to the WSW? Also that bounds to the west the Norcia basin?*

Authors: we agree with this comment. Among the steep normal faults, the WSW dipping faults are not the unique characterizing the area, but the ones that generally produce the stronger earthquakes and therefore are better known with respect to the ENE dipping faults. However, the antithetic ENE dipping faults are also important in this structural context, because they seem to be able to produce moderate earthquakes (as highlighted e.g. by Chiaraluce et al., 2017 in this seismic sequence). The fault that bounds the west side of the Norcia basin, that we clearly recognize in this work, belongs to the second type (ENE dip). We have improved the text following the Rev2 suggestion.

Line 139: … are the Castelluccio di Norcia (CNb) and Norcia (Nb) basins …

*Rev2: Refer to fig 1*

Authors: reference to Fig.1 added.

Lines 140-141: … They have been subjected to a lacustrine and fluvial sedimentation of hundreds of meters …

*Rev2: rewrite*

Authors: we have rewritten the paragraph.

Lines 143-149: … The recent 2016-2017 seismic sequence has been caused by the activation of a complex NNW-SSE trending fault system, characterized by prevalent high-angle WSW-dipping normal faults (Lavecchia et al., 2016). More in detail, the easternmost fault system of the region recently activated is the NNW-SSE trending "Monte Vettore fault system" (Vf). This was the responsible of the mainshock nucleation between the continental Norcia (Nb) and Castelluccio di Norcia basins (CNb) (Fig. 1). Nb and CNb are two asymmetrical grabens, bordered by high-angle WSW-dipping normal faults located on their eastern flanks. Both fault systems are thought to have high seismogenic potential and able to generate earthquakes up to Mw 7.0 …

*Rev2 (Grouped questions): Rewrite. You repeat the same idea in different ways. You have defined acronyms in line 139, use them. This must be located after line 142, when you are describing the basins.*

Authors: corrected and rewritten.

Line 151: … The Nb master fault (Nottoria-Preci fault, Nf) …

*Rev2: Localize in Fig1*

Authors: corrected

Line 158: … Norcia and Castelluccio faults …

*Rev2: Which fault is this one? Localize in Fig1*

Authors: we refer always to the same fault systems mentioned so far, including the synthetic and antithetic ones and their secondary splays. Therefore, we have integrated the text and figures.

Line 171: … whilst explosive was used for NOR02; …

*Rev2: In Table 1 you mention that CAS01 was acquired with explosives. Which ones is correct?*

Authors: text ok, we have updated the table.

Line 174: … parameters in Table 1s, supporting information …

*Rev2: This is in Table 1. There is no table in supporting information*

Authors: corrected, it is in the manuscript.

Line 175: … Some processing artefacts (A) are visible …

*Rev2: In the corresponding figures, put the A on top of the line identifying the artefact.*

Authors: There is already in the figure a label A on the top of the artefact (yellow dashed line). We have improved the figures and text.

Line 176: … CAS01 (Fig. 1s-a, supporting information) …

*Rev2: There is no Fig 1s-a in supporting information. This profile is not well identified in on of the*

*figures available in the supporting information section. Also the figures in the supporting information*

*section are not identified. Finally, most of the figures in the supporting information section are*

*repetitions of the figures provided in the manuscript and must be deleted if not used for anything.*

Authors: corrected, the figure is the Fig. 3a. The figures in the supporting material are effectively the same. But we added here the high-resolution (PDF) version of each figure, because we noticed an excessive compression and quality reduction of the journal printed-pdf after its creation. So, HR figs were added only to help the reviewers during the revision. In addition, after this revision, we'll use the supporting material to add the attributes images (Figs. 2s, 3s, 4s) without any interpretation and line drawing for comparison with the fig.s 2, 3 and 4.

Line 176: … some seismic events and lineaments …

*Rev2: ? Events? Earthquakes? What do you mean by events?*

Authors: "events" removed and replaced in the text as requested.

Lines 177-178: … seems potentially improvable with a proper choice of seismic attributes type and parameters …

*Rev2: ? Rewrite*

Authors: we have rewritten the sentence as requested.

Line 180: … Ithaca database …

*Rev2: reference*

Authors: reference added

Line 183: …Iside database …

*Rev2: reference*

Authors: reference added

Line 190: … Over the last years, …

*Rev2: Maybe explain briefly what is a seismic attribute?*

Authors: we have integrated the text adding also new references

Line 196: … also using composite multi-attribute displays …

*Rev2: Maybe explain briefly what this means?*

Authors: … we have integrated the sentence… as requested

Line 200: "Energy" (E):

*Rev2: I think that is identified as EN in Figures 2, 3 and 4. Be consistent.*

Authors: corrected

Line 207: lateral variations in seismic events,

*Rev2: What do you mean by seismic events? I have done the same question before.*

Authors: we agree there is confusion with events as earthquake. We have modified the text.

**5. Results**

Line 228: … considerable improvements …

*Rev2: I wouldn't say considerable, at least just looking at figures 2, 3 and 4.*

Authors: We do not agree. Surely the low-quality images in the revision pdf don't show efficiently the improvements, but in comparison to the standard lines displayed in amplitude, there are many details and signal characteristics that are enhanced and that in our opinion improve the data interpretability. However, we have attempted to improve the figures to show the benefits provided by attributes.

Line 235: …200 ms in TWT…

*Rev2: Thickness?*

Authors: the thickness in TWT it's about 200 ms. Considering an average velocity of 6000 m/s for the carbonates, the thickness in meters would be about 600 m.

Lines 236-239: A similar feature showing such a peculiar signature is visible also in CAS01, approximately at the same time interval (Fig. 3a, line location reported on the top insert). But in comparison to NOR01, it appears more discontinuous all along the seismic profile, and in addition it is partially interfering with suspicious processing artefacts (highlighted with yellow dots, labelled as "A", slightly undulated in Fig. 3a whilst horizontal in Fig. 2a ca. at 1 s).

*Rev2: I agree, but you should mentioned that it is masked by the artefact (yellow dotted line). In some places seems that it could be directly related to this artefact. I would point that the most clear area is close to the western end of the profile at about 3s TWT. Al the seismic facies in this area are similar to those shown in profile NOR01.*

Authors: We agree, thank you. We left only the shallower artefact, that is very sharp and clear (see in particular EN and PR attributes) and it seems a copy of the topography.

Line 241: … and beneath the southern termination of Nb (ca. between 11-15 km)…

*Rev2: I agree that is clear in the western part of the profile, but not that clear in the eastern. Needs to indicate fault Nb somewhere in the figure, maybe the upper geological map?*

Authors: Nb is already indicated in Figure above the line in standard amplitude, reported as Norcia basin. We have preferred to use in the figures the entire names of the basin, whilst in the text the acronyms to facilitate the reading. However, we have added Nb on the geological map on the top.

Lines 242-243: H is better enhanced in fig. 3b by EN attribute (blue arrows), and in particular by the EG and PR attributes (Figs. 3c and 3d), that considerably help to better detect and mark its extension and geometry.

*Rev2: To me some of the characteristics that you attribute to horizon H are also related to the observed artefact. If you compare the signal of the upper artefact with the signal of the lower artefact it is not*

*very different. Clearly, in the western end of the profile it is more similar to the results in NOR01, but in the other areas is more arguable. Maybe in the places marked by the blue arrow, but I could point to places related to the upper artefact that have a similar signature (yellow arrows in fig 3 show zones with similar characteristics on the upper artefact than those identified as H in the lower part with a blue arrow). To me it is not clear that you can clearly mark its extension and geometry.*

Authors: As described in the comment above, we left only the shallower artefact.

Line 246: … Nb …

*Rev2:* No Nb in the figure

Authors: we have added Nb on the geological map on the top.

Line 249: this discontinuity propagates down to ca. 2.5 s and intercepts the aforementioned strong reflector H.

*Rev2: To me that is not evident at all. Below the artefact it is almost impossible to distinguish any west dipping lineament.*

Authors: We have improved the red arrows to suggest the W-dipping discontinuity visible in fig.2 EG and even better in PR. There is a different reflectors pattern beneath the basin in comparison to the external (east) part and the high-angle discontinuity is in our opinion clearly visible: it propagates down to the depth level in which we find the reflector H. We have made many efforts to improve the figures in the text and also added a new figure with magnifications of the PR attribute (see Fig. 5s).

Lines 250-252: other similar but minor discontinuities can be also noticed crossing and slightly disrupting the shallower reflectors: those high angle features are efficiently displayed by the EG and PR attributes (Fig. 2c, 2d), whilst in the original line in Fig. 2a cannot be really appreciated.

*Rev2: I would not say efficiently displayed. In fact it is difficult to see anything in that zone, even in Fig 6a,d,e. I question that you would have interpreted anything there without the knowledge of surface geology.*

Authors: we have better declared in the text that the surface geology and structural information has been used to "drive" a first phase of interpretation, at least to detect the extension of the basins and the main faults. However, most of the faults interpreted later on the base of the attributes signature (see the new image in Fig. s5) don't have evidences at surface, apart a couple of splays detected by the paleoseismologists close to the Norcia centre (Galli et al, 2005 and 2019).

In our opinion the aNf is clearly visible thanks to the seismic attributes and it has been detected for the first time in a geophysical data across this basin (it is still debated in literature, as it does not have clear surface evidences). Then, the seismic attributes enhanced in particular the secondary splays close to the surface, visible by following the lateral discontinuity of the quaternary deposits at shallow depth. We hope to have provided here useful elements for better illustrate our interpretation. In addition, we have revised and improved all the images in the manuscript.

Line 253: … by similar geophysical features …

*Rev2: Similar to what?*

Authors: we have added "to ones detected in NOR02 and CAS01"

Line 255: … in Figs. 4b and 4c …

*Rev2: Do you mean Figs. 4c and 4d?*

Authors: yes, thank you. We have rewritten the sentence and in general all the paragraph.

Line 256: W-dipping

*Rev2: Do you mean the west dipping or the east dipping? In the range you indicate there is just the east*

*dipping, the west dipping may begin around km 6 and end at 3-4? I could agree that there is something*

*corresponding to the west dipping lineament but I have more difficulties to interpret the east dipping*

*lineament, mainly in 4c, maybe 4d shows a change in general facies east and west of this lineament. On*

*4c (HR image) seems that the red arrows are pointing to arbitrary places not to places with the same*

*characteristics, and that is confusing.*

Authors: We have rewritten the text better separating within the description the reflectors and the alignments (discontinuities). We have improved the figures adding more accurately other smaller arrows and red semi-transparent polygons to attract the readers' attention on the main lineaments thus simplifying the text comprehension.

Lines 259-260: It crosses the entire profile, rising from about 4 s (West) to ca. 2 s (East), where it intercepts one of the high amplitude events on the eastern end of the seismic line (18-20 km).

*Rev2: I agree that there is a west dipping lineament T that cuts H, but I am not sure it is possilbe to*

*follow that lineament from km 10-11 to the east as the authors interpret. It would be necessary an un-*

*interpreted section.*

Authors: Thank you. Yes, we agree that T cuts H and this discontinuity is basically not visible in the original line, but well enhanced for example in PR attribute. In our opinion T can be traced along almost all the line. To convince the reviewer we have added the figure without green dots in the supplementary material as requested (see figs. 2s, 3s and 4s).

Lines 261: … original line …

*Rev2: Amplitude data?*

Authors: yes thank you, it was a repetition so we have decided to remove the sentence.

Lines 263: …is a much clear visualization of the reflection patterns…

*Rev2: ... Much clear? Maybe there are some improvements for some horizons (H and T) but I am not*

*sure about the ones pointed with red arrows or just for some of them. ...*

Authors: we clearly agree about the improvements for the horizons like H and T. Also, the overall pattern of reflectors is much better. To avoid misunderstanding, with the red arrows we don't indicate reflectors, but only the steep discontinuities of phase/amplitude separating the reflectors, that later we interpret as attributes evidence of fault zones, however usually simplified in seismic interpretation with only one red line. We improved the figures to better help the readers in the interpretation.

Lines 264-265: ... a main high-angle E-dipping discontinuity (red arrows) delimits the NOR02 western sector (ca. 1 km of distance along the line at surface); a ...

*Rev2: Specifically, this is one of the high-angle discontinuities that I am not sure about. I could agree*

*that to the east the seismic facies changes, but I cannot identify a clear lineament in any of the attribute*

*profiles.*

Authors: We think the fact that there is a clear and sharp change of the reflection pattern, as the reviewer also noticed, is already an indication of a lateral discontinuity (or sets of discontinuities). We have introduced the concept that we should rethink the concept of single fault planes with distributed

"fault zones", made by many secondary splays and discontinuities at different scales concentrated in a relatively narrow area. This was one of the reasons on the base of our initial seismic interpretation with the arrows and without a conventional line drawing. Seismic attributes like the Pseudo-Relief are able to clearly enhance also small-scale discontinuities providing an outcrop-like seismic line. We think that regarding this fault, the different reflection patterns as well as the phase discontinuities and truncations of some reflectors, suggest the presence of distributed fault zones. Again, we remark that to better support the readers, we have also added an additional image to the Supporting material (Fig. 5s). We made an additional effort refining the arrows, and introducing a simpler line drawing on the main visible discontinuities better magnified by this figure.

Line 266: (red arrows)

*Rev2: Why you do not identify the different lineaments with a letter and a number? E.g., L1, L2... That would be more easy to localize them in the figures and to refer to them.*

Authors: Thank you for the suggestion, but the main faults are already labelled with Nf, aNf, Vf and aVf on the PR figures. Regarding the minor faults, we have preferred to do not add other labels: as remarked also by Rev1 there are already many acronyms and this may make the reading of the document fragmentary. However, we have improved also the text.

Lines: smaller discontinuities pervasively cross-cut the set of reflectors between 1-4 km bounded by such two main features, producing a densely fragmented reflectors pattern in the middle portion.

*Rev2: With the resolution and quality of the data this is very difficult to see. I could point to zones with similar characteristics (just to the east of the profile). Maybe there is some over-interpretation based on surface geology.*

Authors: As (we hope) better visible in the new image, there are many secondary steep discontinuities, that we have traced giving to the data this peculiar fragmented pattern across the fault zones. We have drawn on the top of the PR images only few faults reported on the geological maps by literature. There are many others interpreted not on the base of the surface geology, but we interpreted just the main secondary ones avoiding possible over-interpretations.

Lines 268-269 and 271: ... Another steep E -dipping feature is visible at higher depth (red arrows at 1-3 s, ca. 7-9 km) beneath ... to a similar structure displayed in a more central portion of NOR02 ...

*Rev2: Again this lineament it is not clear to me. In EG I could say that maybe the zone between both lineaments, the one east and the other west dipping, shows a different facies, but there is not a clear lineament. In fact arrows 1 and 2 mark zones with different lineaments to me (see my annotations on the figure and orange dotted lines).*

*Identify the different lineaments with letters in the figures and refer to them in the text. The way you are doing this is confusing. I thought that you were describing the lineament dipping east in the center of the profile. My previous comment was referring to this lineament.*

Authors: we agree that is less clear than other discontinuities, and in addition, here our interpretation has not been driven by the surface geology, because this discontinuity is quite deep. However, as better shown in the new figure, we cannot avoid to notice the E-dipping lineament highlighted in correspondence to the arrowheads in the area at 2 seconds. About the aVf fault, we have already replied above and suggested to see the new figure 5s in Supplementary. We have reorganized the labels of the main discontinuities in the figures as requested.

Line 272: ... here ...

*Rev2: Here? Where?*

Authors: we have rewritten the text.

Lines 274: characterized by very short and fragmented reflectors bounded by those two steep features of opposite dip.

*Rev2: As said in one of my comments before when I thought you were describing this area, I have difficulties to interpret both lineaments.*

Authors: we hope the revised paper has been improved as well as the images easy to interpret.

Lines275-277: ... of such a main W-dipping alignment also seems to truncate and disrupt both the gently-dipping discontinuity T and the deep reflector H: at approximately 3.2 s, it appears interrupted laterally on its western side (Figs. 4c and 4d) ...

*Rev2: I cannot interpret this lineament so far, but according to your interpretation this lineament dipping to the west seems to die on horizon T (Fig4c). As you have mentioned before, seems that is horizon T that truncates horizon H. Profiles does not have the lateral scale to allow the easily*

*identification of the place the authors are mentioning. In fact I am not sure what reflector/discontinuity*

*they are referring here.*

Authors: The lateral scale is reported on the top profile of Fig.s 2a, 3a, 4a) and now we have modified also adding the scale to the bottom of each one. We also suggest that there is a grid of thin black lines in all the images, vertical for the distance (intervals of 5 km) and horizontal for the travel time (every 1

second).

Regarding the deep area between 5-10 km, it the much complex in the data. It seems that T (low angle)

intercept and interrupt H, but we prefer to present and discuss possible interpretations on which tectonic structure cuts the horizon H. As also suggested by the Reviewer 1, we discuss the relationships with the

Acquasanta thrust (low-angle discontinuity T) is more ambiguous.  We propose two alternative interpretations can be proposed, schematically represented in Fig. 9: a model in which Vf merges into the deep Acquasanta thrust (T), suggesting a negative inversion, and another in Vf cuts and displaces the Acquasanta thrust, following a steeper trajectory (ramp). In both cases the H horizon is truncated, but the relations between the Vettore fault, Vf and the Acquasanta thrust, T, are different.

Lines 279-280: displays clarified the deep geometries of the main reflectors and of the geophysical discontinuities, later

*Rev2: Some of the reflectors/discontinuities may have been highlighted by the attribute processing (H*

*and T) but in general I have some problems to identify the lineaments interpreted by the authors.*

Authors: we are glad that some discontinuities have been detected by the reviewer. Regarding the others, we hope our revision have improved the figures, allowing a better visualization of the reflectors mentioned in the text.

Lines 282-283: overlapped using ODT software  (depth conversion with VPav = 6000 m/s, vertical scale 2x).

*Rev2: This must go on the figure caption*

Authors: moved to caption, as requested.

Lines 285-286: The blue box of Fig.5a is reported in Fig. 5b and 5c

*Rev2: ...corresponds... ...to figures ... To me the reflector corresponding to H is very clear in the original seismic image 5b, maybe more clear than in 5c. Then, I am not sure what the attribute analysis is providing. ...*

Authors: H is relatively clear in the original line only for the short portion in which it shows higher amplitude. Looking more globally the lines, the attributes are not only giving a peculiar signature recognizable also on other lines (like CAS01 and NOR1) contributing to give a better idea of its regional extension, but also its lateral extend in NOR02 is better appreciable enhancing its continuation to the east with a gentle W-dip in comparison to the original line (see figure below).

[Figure]

Lines 288-289: The Fig.5e displays the enhancement obtained plotting the PR attribute ("similarity palette") in transparency on the seismic line in amplitude (SA).

*Rev2: I am not an expert in interpreting this type of datasets, but I am having difficulties to see any enhancement in the data in 5e. I cannot see any lineament.*

Authors: we have improved the figure. The dense steep lineaments (discontinuities) highlighted in 5e produce peculiar seismic facies, that in our opinion can be used to interpret the area as a fault zone in which there is a strongly deformed associated with a main fault.

Lines 291-293: ... The comparison between the multi-display of attributes PR and EG (blue box in Fig. 6a), the original line (Fig. 6b) and the EN+PR plot (Fig.6c) shows the improved signature of the strong reflector H. The black box again reports the original line NOR01 and the version PR+SA, clearly ...boosting the visualization of the high-angle discontinuities.  293

*Rev2: ... I think the attributes maybe highlight the reflector H but I also think that in the original dataset is also quite clear, so talking about improving...*

Authors: we have partially already replied above. We think that the improved images are often self-explicating, improving the interpretability of the data. The alternative way is to not use these seismic data. Clearly, the outcomes here provided may be not dramatic as usually happens in modern high-resolution 3D survey, but any improvements, even if only on some reflectors or on limited area, are welcome considering the uniqueness of these seismic lines and the importance of the study area.

Lines 294-295: ... those are connected with the surface geology and related to the hypocentre location of the main seismic events, that will be discussed more in detail ...

*Rev2:... Not sure at all about this ... In fact, it seems that the authors have been interpreting high-angle lineaments based on surface geology as I have commented before. Some of this interpreted lineaments are quite quastionable, at least I have difficulties to interpret them, but, as mentioned before, I am not an expert in this kind of interpretations.*

Authors: as already replied in other comments, we admit that our interpretation was driven by, but not limited to the surface geology. As already pointed out in comments above, some faults have been interpreted in this way, but then for many others, we used typical elements of a standard seismic interpretation (phase discontinuities, lateral variation in amplitude, offset between reflectors etc..).

Lines 306-307:

*Rev2: This must be in the figure caption.*

Authors: Thank you, part of this sentence has been moved to the figure caption.

Lines 312-322: The steep discontinuities highlighted by the attribute analysis are here interpreted as the seismic signature at depth of complex normal faults mapped at the surface.

*Rev2: As said in various comments I have problems to interpret these steep discontinuities. All this discussion is based on the authors interpretation and since I cannot interpret the same things I cannot support it. But again, I am not a specialist in this type of seismic interpretations.*

Authors: we are confident that after the revision and integrations provided, the interpretability of the seismic features will be now more clear for the readers.

Lines 326-329: belonging to a conjugate tectonic system (Brozzetti & Lavecchia,1994; Lavecchia et al., 1994) and suggested by morphological evidences (Blumetti et al., 1990) and paleoseismological records (Borre et al., 2003). It is a synthetic (W -dipping) high-angle, normal fault bordering the eastern flank of Nb ("Nottoria-Preci fault" – Nf, Calamita et al., 1982; Blumetti et al., 1993; Calamita & Pizzi, 1994).

*Rev2: This is referred to the interpreted discontinuity/fault or to the Nf? Which one? Ok to Nf but the previous sentence it is not clear, so this "It is" is also no clear what you are referring to. Rewrite both sentences.*

Authors: yes, it is referred to the E-dipping (antithetic Norcia Fault -aNf-), currently still debated in literature because not clearly visible in outcrops and only inferred, before this study, by geomorphological evidences and by paleoseismological studies (e.g. Galli et al, 2018; Borre et al., 2003) .We have rewritten the text.

Line 330 and 332: ... red arrows, Figs. 2c, 2 d ... and red arrows between 7-9 km, ca. 1-3 s

*Rev2 (grouped comments): Do you mean 4c and 4d? Red arrows, which ones? There are a lot of red arrows. Again, it is necessary to identify the lineaments by names in the figures and in the text. See my previous comments about lineaments identification.*

Authors: yes, thank you for this correction. We have updated the text and the figures and already replied in previous comments.

Lines 345: and the thrust (T) at about 3.2 s.

*Rev2: Seems strange that the normal fault is cutting the thrust plane. Usually in inversion tectonics, the "new" faults use the slip planes of the previous faults, since it requires less effort to slide along a preexisting plane than to generate a new one. In that case, it seems more plausible that the normal faults would be using the thrust detachments at depth as fault planes and not rupturing them and generating new ones.*

Authors: there are currently different interpretations and models available in literature. Our data do not allow to clarify this point in detail, so, following also the suggestions of the Rev. 1, we have provided two possible interpretations on the relations between the thrust and normal fault. In one case the normal fault cuts the thrust and another case characterized by negative inversion of the pre-existing thrust. We have compared and discussed these two models in the Discussion (chapter 6) and produced a new figure (Fig. 8)

Line 363: high-resolution

*Rev2:?*

Authors: high-resolution images. Corrected.

Lines: However, the attributes aid the seismic interpretation to better display the reflection patterns of interest and provided new and original details on complex tectonic region in Central Italy.

*Rev2: Arguable*

FIGURE 1.

REV2:

For each earthquake indicate the date in which occurred and the depth. Could be also possible to plot seismicity? Above 3.0 or 3.5, to show where the earthquakes are localized. I would suggest to plot more clearly the surface rupture traces on the map.

Authors:

We have added and updated all the information in the Fig.1 as requested.

FIGURE 2.

REV2:

Consider to put the A on top of the line.

Authors:

We have moved A to the left, in a place where it doesn't obscure reflections.

REV2:

Identify with a name each possible lineament (L1, L2,...). The same lineament in two different profiles could have the same name (NOR01 and NOR02).

Authors:

Thank you, we have updated and enhanced the labels for the main faults (aNf, aVf, Nf, Vf) using a continuous line for each one. We didn't add more labels on the interpreted secondary splays but we have added a new figure (fig. 5s) as supplementary material to provide further details on the shallow part of NOR01 and NOR02.

FIGURE 3.

REV2:

Consider to put the A on top of the line.

Authors:

We have moved A to the left in a place where it doesn't obscure reflections, we think it's preferable to maintain the label close to the yellow dots to aid the readers.

FIGURE 4.

REV2:

Identify with a name each possible lineament (L1, L2,...). The same lineament in two different profiles could have the same name (NOR01 and NOR02).

Authors:

Please, see our replies in previous comments.

FIGURE CAPTIONS:

Line 681: Figure 2

*Rev2: For the different figures containing seismic profiles I suggest to explain at the end what means*

*each arrow, dotted line,... Then you avoid repetition or not mentioning in one of the subfigures, as for*

*example not mentioning the red arrows in 2c and mentioning in 2d.*

Authors: fixed following the comment and almost all the captions have been considerably rewritten.

Line 685: ..., with same attributes computation ...

*Rev2: Same as what? A figure caption has to be self explained.*

Authors: deleted